# Fvsoomm a Fuzzy Vectorial Space Model and Method of Personality, Cognitive Dissonance and Emotion in Decision Making †

**Joël Colloc** 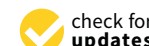

UMR IDEES 6266, University Le Havre Normandy, 76600 Le Havre, France; joel.colloc@univ-lehavre.fr
† This paper is an extended version of our paper published in ESM'2019 EUROSIS Palma de Mallorca Spain: "The evolution of artificial intelligence towards autonomous systems with personality simulation", pp. 61–72.

**Abstract:** The purpose of this extension of the ESM'2019 conference paper is to propose some means to implement an artificial thinking model that simulates human psychological behavior. The first necessary model is the time fuzzy vector space model (TFVS). Traditional fuzzy logic uses fuzzification/defuzzification, fuzzy rules and implication to assess and combine several significant attributes to make deductions. The originality of TFVS is not to be another fuzzy logic model but rather a fuzzy object-oriented model which implements a dynamic object structural, behavior analogy and which encapsulates time fuzzy vectors in the object components and their attributes. The second model is a fuzzy vector space object oriented model and method (FVSOOMM) that describes how-to realize step by step the appropriate TFVS from the ontology class diagram designed with the Unified Modeling Language (UML). The third contribution concerns the cognitive model (Emotion, Personality, Interactions, Knowledge (Connaissance) and Experience) EPICE the layers of which are necessary to design the features of the artificial thinking model (ATM). The findings are that the TFVS model provides the appropriate time modelling tools to design and implement the layers of the EPICE model and thus the cognitive pyramids of the ATM. In practice, the emotion of cognitive dissonance during buying decisions is proposed and a game addiction application depicts the gamer decision process implementation with TFVS and finite state automata. Future works propose a platform to automate the implementation of TFVS according to the steps of the FVSOOMM method. An application is a case-based reasoning temporal approach based on TFVS and on dynamic distances computing between time resultant vectors in order to assess and compare similar objects' evolution. The originality of this work is to provide models, tools and a method to design and implement some features of an artificial thinking model.

**Keywords:** time fuzzy vectorial space; finite state automata; psychism; personality; cognitive dissonance; emotion; neurosis; pathological decision-making; gaming addiction; artificial thinking model; fuzzy object-oriented design method

---

## 1. Introduction

The simulation of the ability to think and to equip computers with an artificial psyche, a personality taking into account pathologies and emotions, represents significant challenges. This work is an extension of the proposed article at the ESM'2019 conference [1]. It complements the feasibility, models, methods and tools available to build such systems.

The challenge is the modelling of an artificial thinking model (ATM) based on perceptions of the characteristics of the world's objects, the emotions these objects provoke and how they interact with and impact on our personality and our ability to grasp new situations and to model in a feedback loop

our way of thinking. In this article, we first propose a state of the art of fuzzy logic with its advantages and disadvantages to represent emotional states and a psychic model.

The Materials and Methods section proposes four contributions:

The first describes the temporal fuzzy vector space (TFVS) model which is not a fuzzy logic approach but rather a fuzzy extension of object models which integrates fuzzy characteristic functions to object-attribute relationships (has-a) and composition (is-part-of) using temporal vectors. This time-object-oriented model allows attributes and objects to evolve and modulate their importance over time. It complements the semantic abilities of object models by providing the opportunity to express their evolution and behavior.

These extensions are necessary for the implementation of a dynamic mental model based on the cognitive model. The second contribution is the fuzzy vector space object oriented model and method (FVSOOMM) which describes the different steps necessary to build the temporal fuzzy vector space (TFVS) from the class diagram describing the knowledge ontology of the system in unified modeling language (UML) [2]. The third contribution describes the cognitive model Emotion, Personality, Interaction, Knowledge *(connaissance)*, Experience (EPICE) and its different layers to describe the elements of an artificial psyche. The fourth contribution is the artificial thinking model (ATM) presented in the conference paper [1] which is based on a double pyramid: the cognitive pyramid that includes two levels, a symbolic level and a subsymbolic level and the linguistic pyramid based on language abilities and a linguistic conception of thought. The results section presents the notion of cognitive dissonance and two examples of the application of the TFVS model—the modelling of emotion during the purchase decision and the application to gambling addiction. The discussion section shows the benefits of the proposed approaches in computer sciences and psychological modelling and the research opportunities to improve them in the future work section that includes the development of a platform to automate the steps of the TFVS design method.

The conclusion summarises the objectives of this work and with humility the results obtained, and recalls the ethical aspects which the artificial intelligence researcher must keep in mind, in particular in respect of users' wishes and freedom.

## 2. State of the Art of Knowledge and Time Modeling with Fuzzy Logic

Since the fuzzy set theory was introduced by Lofti Zadeh, a lot of models was proposed to offer many useful applications. Most of the time, the models relies on a logic approach based on a generalised modus ponens using fuzzy rules [3–6]. These fuzzy rules combine *membership functions* and *linguistics variables* to describe subsets of characteristics values of specific attributes or parameters used to express knowledge chunks stored in knowledge bases. The fuzzy set theory constituted a great enhancement and offered a better flexibility to take into account the uncertainty and the approximation of necessary variable features.

The main drawback of fuzzy rules comes from the boolean operating nature of inference engines that combine rules by firing them or not. To enhance this process, a lot of accurate defuzzification methods was proposed to allow fuzzy rules to be triggered by inference engines, for example [7,8]. The whole process of fuzzification/defuzzification with a Mamdani controller is described in Reference [9] and shown in Figure 1. However, these applications are using a low number of attributes [10] and they are working well because they operate in closed physical systems [11]. One attribute already needs to combine several membership functions with a triangular shape to represent the different threshold values. Trapezoidal functions are used when it is necessary to express a range of values where the function is increasing, a range of values where it is constant and a range of values where it is decreasing. They are exploited by fuzzy logic implication as proposed by Lukasievicz, Mamdani, Larsen [12–18].

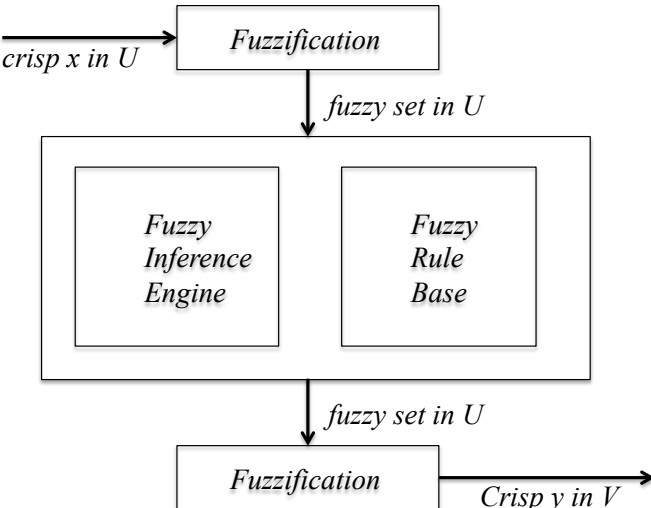

**Figure 1.** A fuzzy logic controller according to Lee [19].

For example, the Figure 2 shows the different classes of the attribute person.size in centimeters and their membership functions VerySmall, Small, Medium, Tall, VeryTall. That requires to implement the corresponding rules. These approaches quickly become very complex as soon as a large number of attributes and objects are involved in the system. Most of the time, the thresholds are defined by experts in an empirical way and with an uncertainty and they are only able to tell that a parameter or an attribute is important to assess a situation but more rarely to precise a specific threshold value, if it is scientifically available. Moreover, in empirical domains like psychology and more generally in human sciences, applications involve a lot of attributes of many objects that interact over the time. All the improvements provided by fuzzy membership functions are almost annihilated by the fact that rules are firing or not, through a boolean process by the inference engine. Besides the adequate implementation of fuzzy membership functions requires to multiply the number of rules and to use a fuzzification/defuzzification process for each of them. Moreover, another conceptual drawback is that inference rules implement a succession of deductions corresponding to a list of object characteristics rather than comparing them globally. According to cognitive psychology, the most used way of reasoning is analogy rather than deduction. Our approach brings the following advantages: It is much simpler, each attribute is represented by only one fuzzy membership function without threshold (and its contrary if necessary). Each simple object can combine many attributes and is represented by one resultant vector. Each complex object of the ontology is composed of other objects and in turn is represented by a resultant vector. Our model benefits from the semantics of object-oriented models and it allows the application of the fuzzy paradigm not only to attributes but also to simple objects and compound objects. This allows analog reasoning at different levels of the object composition hierarchy. In previous works, we have shown that object-oriented approaches are well suited to building very complex ontologies in infectious disease diagnosis and antibiotic prescription, which needs to assess a lot of diseases, drugs, clinical objects and attributes [20,21]. These previous works were not using the TFVS model. We are following Lofti Zadeh himself when he writes in a recent paper, *"Fuzzy logic is much more than a logical system. It has many facets. The principal facets are: logical, fuzzy-set-theoretic, epistemic and relational. Most of the practical applications of fuzzy logic are associated with its relational facet. In this paper, fuzzy logic is viewed in a nonstandard perspective. In this perspective, the cornerstones of fuzzy logic – and its principal distinguishing features – are: graduation, granulation, precisiation and the concept of a generalized constraint."* [22]. In the same way we propose in the following section a fuzzy object-oriented extension which offers fuzzy relationships to capture more semantics and especially time modelling.

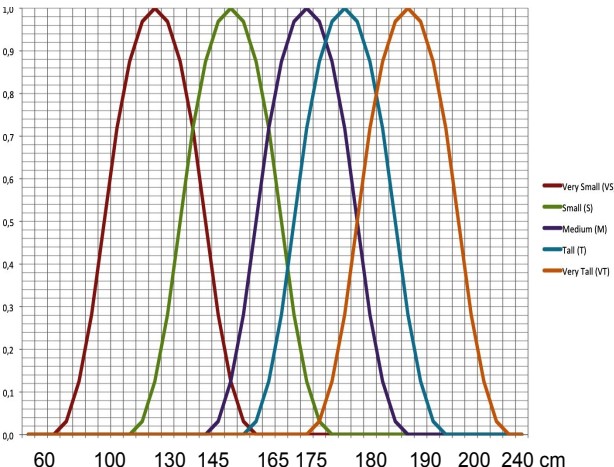

**Figure 2.** Bell membership functions representing the different size classes in centimeters.

## 3. Materials and Methods

### 3.1. Object Oriented Time Fuzzy Vectorial Space

This section proposes a model to describe the emotion and the personality layers of the EPICE model presented in the previous sections. The fuzzy vectorial space as an extension of Zadeh's fuzzy logic [23–25]. Unlike the Zadeh's model which makes it possible to build fuzzy production rules and develop a reasoning possibility based on fuzzy implication and the generalised modus ponens as proposed by Lukasiewicz, [17], Mamdani [12] and Larsen. Our fuzzy vectorial space model is integrated with the composition operator of the object-oriented model [26] and the concept of dynamical functions on composition links and the object-attributes (has-a) relationships [27]. Zadeh's membership functions are defined on [0,1]. Our membership functions are defined on $[-1, 1]$, taking into account a full specific characteristic of an object (value = 1) and its contrary (value = $-1$), where the value 0 is considered to be neutral. The set $\mathbb{F}$ contains all the available membership functions. Thus $\forall f \in \mathbb{F} \exists f' = -f$ defined in $[-1, 1]$ which represents the opposite function of the f function. Let there be three functions of $\mathbb{F}$, $f : [-1, 1] \rightarrow [-1, 1], x \rightarrow f(x), g : [-1, 1] \rightarrow [-1, 1], y \rightarrow g(y), h : [-1, 1] \rightarrow [-1, 1], z \rightarrow h(z)$. We propose a fuzzy vector space among which vectors $\vec{f}, \vec{g}, \vec{h}$ express forces (as a physical metaphor) having as *mode* the "intensity" based on the value of the fuzzy membership functions $f(x), g(y), h(z)$ related to the values $x, y, z$ corresponding with specific properties (attribute, or characteristics) of an object of the domain of the discourse. $\mathbb{R}$ is a commutative corpus. Let E be a vector space on $\mathbb{R}^3$ associated with an orthonormal coordinate system $(O, \vec{i}, \vec{j}, \vec{k})$ from the set of membership functions $\mathbb{F}$. The vector $\vec{f}$ overlaps the vector $\vec{i}$ (X axis), $\vec{g}$ the vector $\vec{j}$ (Y axis) and $\vec{h}$ the vector $\vec{k}$ (Z axis). The norm of the vector $\vec{f}$ is given by $\|\vec{f}\| = \sqrt{f(x)^2} = f(x)$ and in the same way $\|\vec{g}\| = \sqrt{g(y)^2} = g(y)$ and $\vec{h}$ est $\|\vec{h}\| = \sqrt{h(z)^2} = g(z)$. Because the system is orthonormal the dot product (or scalar product) of vectors $\vec{f}$, $\vec{g}$ and $\vec{h}$ is nil. Consequently, we can combine three forces represented by vectors $\vec{f}$, $\vec{g}$ and $\vec{h}$ with an inner additive operator named $+$. $\forall \vec{f}, \vec{g}, \vec{h} \in E$. A vector space E possesses the following properties—associativity of $+$ , neutral element ($\vec{0}$), the opposite vector, commutativity of $+$ in E, scalars, neutral element, distributivity of scalars on addition.

p1:   $\vec{u} = (\vec{f} + \vec{g}) + \vec{h} = \vec{f} + (\vec{g} + \vec{h})$ associativity.
p2:   $\vec{0} + \vec{f} = \vec{f} + \vec{0}$ the neutral element is the vector $\vec{0}$.
p3:   $\forall \vec{f} \in E, \exists -\vec{f} / \vec{f} + -\vec{f} = \vec{0}$ the opposite vector.
p4:   $\vec{u} = \vec{f} + \vec{g} = \vec{g} + \vec{f}$ Commutativity of + in E.
p5:   $\forall \vec{f} \in E, \exists \lambda, \mu \in \mathbb{R} / \lambda(\mu \vec{f}) = (\lambda \mu)\vec{f}$, $\lambda$ and $\mu$ are scalars
p6:   $\forall \vec{f} \in E, \exists e = 1 / e\vec{f} = \vec{f}e = \vec{f}$ Neutral element.
p7:   $\forall \vec{f} \in E, \forall \lambda, \mu \in \mathbb{R}, (\lambda \mu)\vec{f} = \lambda\vec{f} + \mu\vec{f}$, the result is generally outside of $[-1, 1]$.
p8:   $\forall \vec{f}\vec{g} \in E, \forall \lambda \in \mathbb{R}, \lambda(\vec{f} + \vec{g}) = \lambda\vec{f} + \lambda\vec{g}$, the result is generally outside of $[-1, 1]$.

Then E is a vector space on $\mathbb{R}^3$. Let $\mathbb{R}$ be the commutative corpus provided with the absolute value and *E* the $\mathbb{R}$-vector space previously defined, a norm on *E* is an application *N* on E with positive real numbers satisfying the following properties:

- Separation: $\forall \vec{f} \in E, N(\vec{f}) = 0 \Rightarrow \vec{f} = \vec{0}_E$;
- Homogeneity: $\forall (\lambda \vec{f} \in \mathbb{R} \times E, N(\lambda \vec{f}) = |\lambda| N(\vec{f})$.
- Sub-additivity or triangular inequality: $\forall (\vec{f}, \vec{g}) \in E^2, N(\vec{f}, \vec{g})^2 N(\vec{f})$.

### 3.1.1. Calculation of the Resultant Vector of Three Fuzzy Forces

Let *f* be a membership function $f : [0,1] \rightarrow [0,1] / f(x) = \dfrac{1}{1 + e^{-k(x-s)}}$ which is defined and continuous and where *x* is a specific characteristic of an object, *s* is the threshold that expresses the median value for the parameter *x* and *k* is a constant expressing the precision with which *x* is known. To adapt this function to the interval $[-1,1]$, a change of coordinate system and scale is necessary: $F(x) = 2f(x) - 1$ gives $F(x) = \dfrac{2}{1 + e^{-k(x-s)}} - 1$. *F(x)* is an odd function defined and continuous on $[-1,1]$, thus on intervals $[-1,0]$ et [0,1] having 0 for intersection. *F(x)* allows us to build two opposite functions $f : [0,1] \rightarrow [0,1], f(x) = \dfrac{2}{1 + e^{-k(x-s)}} - 1$ and $f' : [-1,0] \rightarrow [-1,0] \, f'(x) = \dfrac{2}{1 + e^{-k(x-s)}} - 1$. It follows that $\forall x \in [-1,1], f(x) = -f'(-x)$. In the same manner, we define $G(y), g(y), g'(y)$ and $H(Z), h(z), h'(z)$. Note that *x* and also the parameters *y* and *z* correspond to different relevant characteristics of the studied object. Then according to the sign of a characteristic *x*, it is always possible to build the corresponding membership function *f* and its opposite function *f'* that expresses the contrary characteristic $-x$ (for example: *x*: to be tall corresponds to the opposite $-x$: to be small and *y*: to be joyful, $-y$: to be sad). The vector $\vec{f}$ is defined from *f(x)* and the opposite vector $\vec{f'}$ is defined from *f'(x)* and respectively vectors $\vec{g}$ and $\vec{g'}$ from functions *g(y)* and *g'(y)* and vectors $\vec{h}$ and $\vec{h'}$ from functions *h(z)* and *h'(z)*. This adaptation allows to satisfy the property of having an opposite vector: $\forall \vec{f} \in E, \exists -\vec{f} / \vec{f} + -\vec{f} = \vec{0}$ that is necessary to build a fuzzy vector space.

### 3.1.2. Properties of *f(x)*, *g(y)* and *h(z)*

Let *E* be the vector space on $\mathbb{R}^3$, $E(O, \vec{i}, \vec{j}, \vec{k})$, the resultant vector is:

$$\vec{u} = \vec{f} + \vec{g} + \vec{h}. \|\vec{u}\| = \sqrt{f(x)^2 + g(y)^2 + h(z)^2}. \tag{1}$$

Because $f(x), g(y), h(z) \in [0,1], \max(\|\vec{u}\|) = \sqrt{3}, \|\vec{u}\| \in [0, \sqrt{3}]$. The resultant function *r* of the membership functions *f(x)*, *g(y)* and *h(z)* is:

$$\forall (x,y,z) \in [0,1]^3, r(x,y,z) = \frac{1}{\sqrt{3}} \|\vec{u}\| = \frac{1}{\sqrt{3}} \sqrt{f(x)^2 + g(y)^2 + h(z)^2}, r(x,y,z) \in [0,1]. \tag{2}$$

In the vector space E, we can associate a scalar with a vector that expresses the importance of the component in the linear combination of vectors. $\alpha, \beta, \gamma \in R^{+*}, \vec{u} = \alpha \vec{f} + \beta \vec{g} + \gamma \vec{h}$ and the norm becomes

$$\|\vec{u}\| = \sqrt{\frac{|\alpha| f(x)^2 + |\beta| g(y)^2 + |\gamma| h(z)^2}{|\alpha| + |\beta| + |\gamma|}}. \tag{3}$$

Regardless of the use of scalars, we verified that $r(x,y,z) \approx \frac{1}{\sqrt{3}} \|\vec{u}\|$. The study of the properties of the opposite functions $f'(x), g'(y)$ and $h'(z)$ defined on $[-1,0]$ is similar. Because each function *f(x)* comes with its opposite function *f'(x)*, $\alpha f(x) = -\alpha f'(-x)$, the sign of $\alpha$ allows us to set up whether the function *f* is agonist or antagonist and in the same manner $\beta$ for *g(y)* and $\gamma$ for *h(z)*. Consequently, positive values and negative values must be computed separately.

### 3.1.3. Generalization in n Parameters

The model can be adapted to take into account n parameters or characteristics of objects $n \ll \infty$, $x_i \in [0, 1]$ and n membership functions $f_i(x_i)$ with a scalar $\alpha_i \in \mathbb{R}^n$. A vector space E is defined on $\mathbb{R}^n$ where two vectors $\vec{f} = (x_1 \ldots x_n)$ and $\vec{g} = (y_1 \ldots y_n)$ have the inner scalar product (dot product) $\langle \vec{f}, \vec{g} \rangle = x_1 y_1 + \ldots + x_n y_n$. E is provided with an orthonormal coordinate system $O(\vec{i_i}, \ldots, \vec{i_n})$. Let be the vector $\vec{u}$ in the vector space E on $\mathbb{R}^n$ to be the sum of vectors $\vec{f_i}$ provided with the corresponding scalar $\alpha_i$. Thus comes $\forall \vec{f_i} \in E, \vec{u} = \sum_{i=1}^{i=n} \alpha_i \vec{f_i}$. The norm:

$$\|\vec{u}\| = \sqrt{\frac{\sum_{i=1}^{i=n} |\alpha_i| (f_i(x_i))^2}{\sum_{i=1}^{i=n} |\alpha_i|}}. \tag{4}$$

The resultant function r of n membership functions is:

$$\forall (x_i, \ldots, x_n) \in [-1, 1]^n, r(x_i, \ldots, x_n) = \frac{1}{\sqrt{n}} \sqrt{\frac{\sum_{i=1}^{i=n} |\alpha_i| (f_i(x_i))^2}{\sum_{i=1}^{i=n} |\alpha_i|}}. \tag{5}$$

The main inconvenience is when n becomes big, the value of the resultant function r becomes very tiny. Indeed, $\lim_{n \to +\infty} r(x_i, \ldots, x_n) = 0$. However, this inconvenience is not important because in practice the number n of membership functions $n \ll \infty$ and is some tens of parameters at most. The Fuzzy Vectorial Space is a new way of combining efficiently fuzzy vectors without using fuzzification/defuzzification steps as in [12] and it maintains all the benefits of the fuzzy sets proposed by Zadeh as well as to take time into account as described in the next section.

### 3.1.4. Consideration of Time

We saw that the evolution of the characteristic parameters of objects requires the consideration of time. During time t, parameters can evolve quickly or slowly, or sometimes remain constant. The resultant vector $\vec{u}$ undergoes modifications during time t. The absolute time is expressed by an additional variable t (in seconds) which is calculated from 6 variables—year (aaaa), month (mm), day (jj), the hour (hh), minutes (mm), seconds (ss). An extension in the subdivisions of second may be envisaged for special applications but is mostly useless and expensive in computation time. The other temporal units (for example, week, month, half-year) associated with the various parameters can easily be converted in absolute time by appropriate conversion functions. Most programming languages offer such time function libraries. The variable t is used to define every membership function $f_i(x_i)$ which becomes $f_i(x_i, t) : [0, 1] \to [0, 1], f_i(x_i, t) = \dfrac{2}{1 + e^{-k(x_t - s)}} - 1$. Some functions remain constant on an interval of time while others are evolving. For all time t, the vector $\vec{u_t}$ evolves according to its component functions: $\forall \vec{f_{i,t}} \in E, \vec{u_t} = \sum_{i=1}^{i=n} \alpha_{i,t} \vec{f_{i,t}}$. The norm is :

$$\|\vec{u_t}\| = \sqrt{\frac{\sum_{i=1}^{i=n} |\alpha_{i,t}| (f_{i,t}(x_{i,t}))^2}{\sum_{i=1}^{i=n} |\alpha_{i,t}|}}. \tag{6}$$

The resultant function r of n membership functions is:

$$\forall (x_{i,t}, \ldots, x_{n,t}) \in [-1, 1]^n, r(x_{i,t}, \ldots, x_{n,t}) = \frac{1}{\sqrt{n}} \sqrt{\frac{\sum_{i=1}^{n} |\alpha_{i,t}| (f_{i,t}(x_{i,t}))^2}{\sum_{i=1}^{i=n} |\alpha_{i,t}|}}, r(x_{i,t}, \ldots, x_{n,t}) \in [-1, 1]. \tag{7}$$

### 3.1.5. Kinematic of a Point in the Fuzzy Vector Space

This part is inspired from physics and more precisely from classical mechanics. Therefore, we use a three-dimensional cartesian vector space to illustrate our proposal. Let time t be in the interval $[t_0, t_n]$ and the vector space E provided with the orthogonal coordinate system $(O, \vec{i}, \vec{j}, \vec{k})$. The vector

function $\vec{U}(t)$ in E is defined as $\vec{U}(t) = F(x,t)\vec{i} + G(y,t)\vec{j} + H(z,t)\vec{k}$. As previously, $x_t, y_t, z_t$ are the characteristics of an object at the moment t and the components are defined as $x_t, y_t, z_t \in [0,1]^3, t \in [t_0, t_n], \vec{F}(t) = F(x_t, t) \cdot \vec{i}, \vec{G}(t) = G(y_t, t) \cdot \vec{j}$ and $\vec{H}(t) = H(z_t, t) \cdot \vec{k}$ of $\vec{U}(t)$. All of them are fuzzy membership functions of time. According to our previous definition of fuzzy forces in a vector space E, that is the case. Indeed, $F(t) : [-1,1] \to [-1,1], F(t) = \dfrac{2}{1 + e^{-k(x_t - s)}} - 1$ and G(t) and H(t) are defined and continuous in the same way. So $\vec{U}(t)$ is a continuous vector function. O is the origin of the orthogonal coordinate system of the vector space where a point $M = (F(x_t, t), G(y_t, t), H(z_t, t)) = F(x_t, t)\vec{i} + G(y_t, t)\vec{j} + H(z_t, t)\vec{k} / \vec{OM} = \vec{U}(t)$ depicts a curve which is called the hodograph of the vector function $\vec{U}(t)$ which is the position vector at time t in the fuzzy vector space E. An example of the trajectory of the M point in the fuzzy vector space E shown in Figure 3 where the right-hand rule orientation is used.

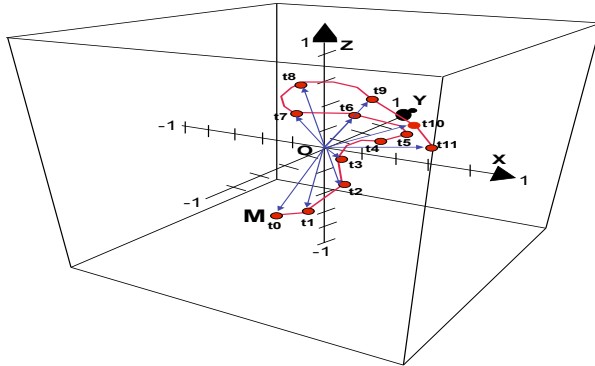

**Figure 3.** Hodograph of the vector function $\vec{U}(t)$ and dot M in the fuzzy vector space E.

### 3.1.6. Velocity and Speed in the Vector Space

The derivative of $\vec{U}(t)$ is $\dfrac{d\vec{U}}{dt} = \lim_{t \to t'} = \dfrac{\vec{U}(t) - \vec{U}(t')}{t - t'}$. $\dfrac{d\vec{U}}{dt}$ exists, if its components $\vec{F}(t), \vec{G}(t)$ and $\vec{H}(t)$ denote the derivatives $\dfrac{d\vec{F}(t)}{dt}, \dfrac{d\vec{G}(t)}{dt}$ and $\dfrac{d\vec{H}(t)}{dt}$. The average velocity becomes the derivative of the position vector $\dfrac{d\vec{U}(t)}{dt} = \dfrac{d\vec{F}(t)}{dt} + \dfrac{d\vec{G}(t)}{dt} + \dfrac{d\vec{H}(t)}{dt}$. Thus, the velocity is the time rate of change of position in the vector space E and $\dfrac{d\vec{F}(t)}{dt}, \dfrac{d\vec{G}(t)}{dt}, \dfrac{d\vec{H}(t)}{dt}$ denote the derivative in the three components, so called dimensions with respect to time. The speed S of the point M is defined as the magnitude $S = |\vec{U}(t)| = \dfrac{ds}{dt}$ where s is the arc-length measured along the trajectory of the point M in the vector space E which is a non-decreasing quantity. Therefore $\dfrac{ds}{dt}$ is non-negative, which implies that the speed S is also positive or zero.

### 3.1.7. Acceleration

The acceleration A of M relies on the rate of change of the velocity vector $\dfrac{d\vec{U}}{dt}$ so, $A = \dfrac{d^2\vec{U}}{dt^2}$. The acceleration is the first derivative of the velocity vector $\dfrac{d\vec{U}}{dt}$ and the second derivative of the position vector $\vec{U}(t)$. In the following section we present an example of application of our model to build an emotion fuzzy vector space.

### 3.1.8. Derivative of the Sigmoid Fuzzy Membership Function

Each component $\vec{F}(t), \vec{G}(t), \vec{H}(t)$ of the vector $\vec{U}(t)$ is defined by the same sigmoid membership function:

$$\vec{F}(t) = F(x_t, t) = \frac{2}{1 + e^{-k(x_t - s)}} - 1.$$

The derivative of the sigmoid membershipfunction is calculated according to

$$x_t, \frac{dF(x_t, t)}{dt} = \frac{2k \cdot e^{-k(x_t - s)}}{(1 + e^{-k(x_t - s)})^2}. \tag{8}$$

Provided that k is a positive constant, we notice that the derivative $F'(x_t, t)$ is strictly positive. Therefore, $F(x_t, t)$ is always derivable and strictly increasing on $\mathbb{R}$. Obviously, the same properties are verified for the functions $G(y_t, t)$ and $H(z_t, t)$. The second derivative is:

$$F''(x_t) = \frac{2 \cdot k \cdot e^{(-k \cdot (x_t - s))}}{(1 + e^{(-k \cdot (x_t - s))})^2}. \tag{9}$$

The third derivative is:

$$F'''(x_t) = \frac{-2 \cdot k^2 \cdot e^{(-k \cdot (x_t - s))} + 2 \cdot k^2 \cdot e^{(-k \cdot (x_t - s))^3}}{(1 + e^{(-k \cdot (x_t - s))})^4}. \tag{10}$$

The third derivative on Figure 4 is the *"jerk function"* which indicates a bifurcation of the membership function and thus a radical change of the evolution of the corresponding attribute value at this time. The Figure 5 describes how the TFVS of attributes and the is-part-of composition relationship are combined together to build a resultant fuzzy vector according to Section 3.1.

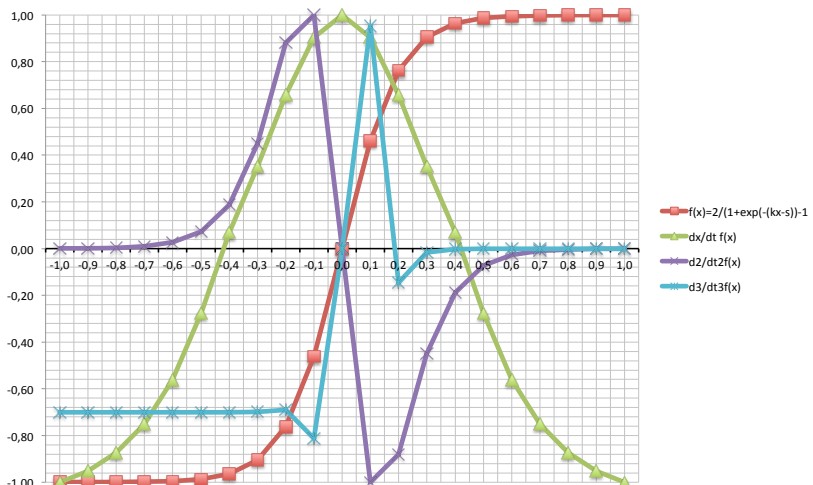

**Figure 4.** First, second and third derivative of the sigmoid membership function.

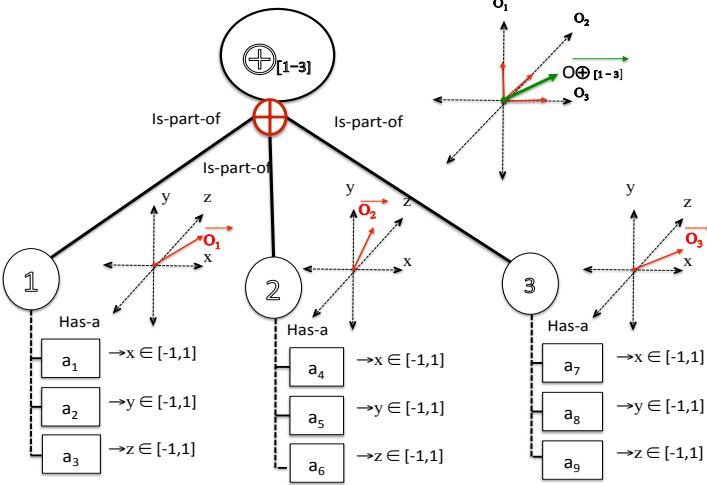

**Figure 5.** Fuzzy vectorial space of a composite object with their attributes.

## 3.2. Method to Implement Fuzzy Vector Space in an Object Oriented Model

A method provides guidelines and procedures to design an information system. Methods are mainly inspired by IT systemic methods like Merise [28]. Some methods are more oriented around artificial intelligence (AI) design like *"Knowledge Oriented Design"* (KOD) by Claude Vogel, *"Knowledge Analysis and Design Support"* (KADS) from Schreiber, Wielinga, and Breuker [29], and more recently the systemic method *"Méthode d'Analyse et de Structuration des (K)Connaissances"* (MASK) proposed by Jean-Louis Ermine. This section will present briefly a method to use TFVS to model knowledge with an object oriented approach and the use of the composition operator and the (object "has-a" attribute) relationship [30]. The Figure 6 describes the different necessary steps to design an AI decision support system with time modelling from an object-oriented UML class diagram.

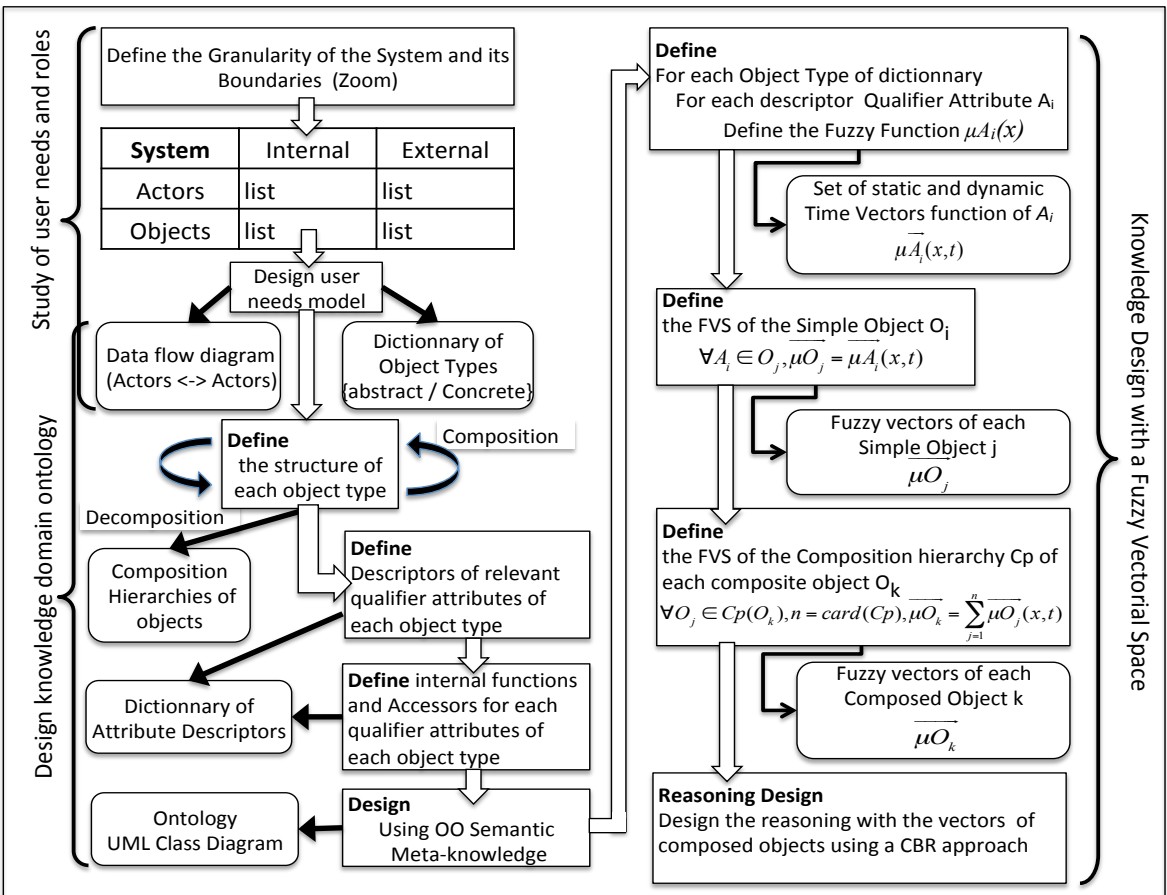

**Figure 6.** A method to design artificial intelligence (AI) decision support systems with fuzzy vectorial space.

The first step concerns *"The study of users needs and roles"* which defines the sub-system to be described and the granularity and its internal/external boundary. Then it is now possible to list the internal and external actors and the roles of each of them and all the relevant objects that are used by the actors. We can then design the data flow diagram and the dictionnary of the object types.

The second step: *"Design knowledge domain ontology"* allows to define all the object types (or classes), their composition hierarchy (is part of) and all the relevant qualifier attributes attached to the objects of this type. A qualifier attribute is a necessary attribute for assessing the state of the object type. Identifiers are unique, not null qualifier attributes that allow to identify an object of the system. Then, as usual in object-oriented models, it is important to define relationships between object types and mainly the generalization/specialization (is-a), the composition (is-part-of) relationships and other specific associations that can provide new object types and specific qualifier attributes. Then, for each

object type, it is necessary to define private internal functions and public accessors. The object types described in the ontology can be mapped in one or several UML class diagrams provided that the semantic relationships of the domain ontology are well respected as in the ontology of the diagnosis of infectious diseases [21]. The third step concerns the knowledge design with a fuzzy vector space and relies on the composition relationship and the qualifier attribute descriptors that defines the characteristics of each attribute of the object type and thus of the object instances Table 1.

**Table 1.** Qualifier attribute descriptor.

| Fields | Description |
| --- | --- |
| *a*Id | Attribute identifier |
| *a*Name | unique descriptive name chosen by the expert |
| aType | Type of value {Boolean, Character, Integer, Float, Double, String, DateHour, Enum} |
| aW | Weight of the attribute in the object $\in [0, 1] or \in [0, 100\%]$ |
| aV | Current value of the attribute |
| aU | Unit of the attribute if any (optional) |
| aVmin | Minimum value of the attribute |
| aVmax | Maximum value of the attribute |
| aConst | The value of the attribute is constant |
| aMand | The value of the attribute is mandatory (may be 0) |
| aLinLog | Linear, logarithmic scale $Log_{base}$ |
| aFuzzy | Fuzzy value computed by the characteristic fuzzy function |
| aAccess | attribute accessors (get, set, modify) are functions that control its update |

It is necessary to sort the static and dynamic attributes. The Fuzzy vectorial space of each object is built bottom-up along the composition hierarchy—beginning from the simple objects that have only attributes and the composite objects whose they are part-of their composition hierarchy as in Figure 5. For each qualifier attribute a sigmoid fuzzy membership function is defined like Figures 7 and 8 in Section 4.2 and thus the corresponding vectors. The last step is to design the reasoning using the vectors of the composed objects, which allows us to recognize complex situations and take appropriate decision. The available reasoning modes are deduction, induction, abduction, analogy, subsumption and case-based reasoning briefly described in Section 3.3.4. The deduction can be implemented at a more macroscopic level by comparing object states (described by their resultant vectors) and taking into account time. A future work will present a methodology to describe how to use the TFVS with each reasoning mode and enhance the knowledge (Connaissance) layer of the EPICE model presented in the next section.

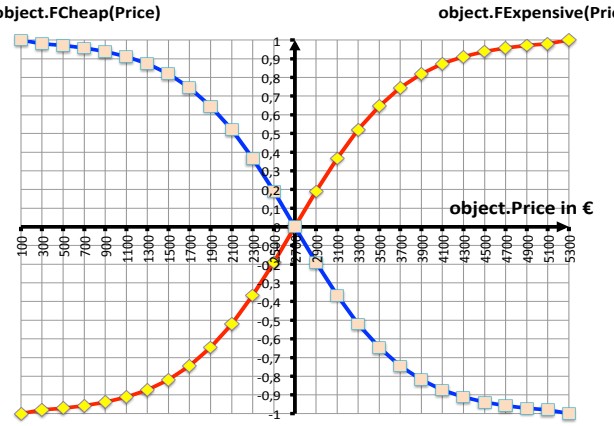

**Figure 7.** Price Membership functions *FCheap()* and *FExpensive()*.

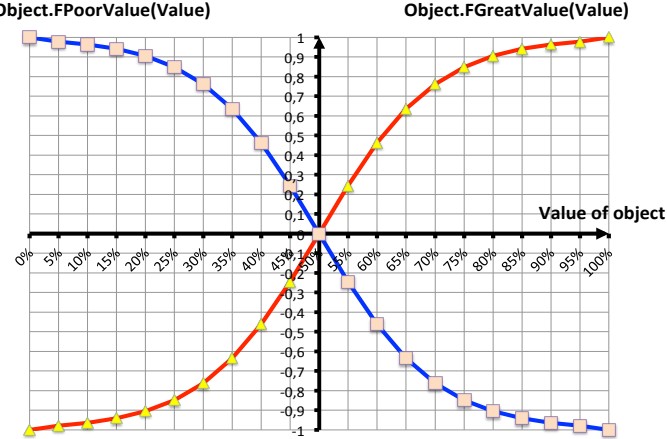

**Figure 8.** Value of the object membership functions: *FpoorValue() and FgreatValue()*.

### 3.3. The EPICE Model

The EPICE model described on Figure 9 according to the French acronym (Emotions, Personnalité, Interaction, Connaissances, Expérience) describes the layers (Emotions, Personality, Interaction, Knowledge, Experience) and implements them in a decision support system in medical ethics [31].

The implementation is done with the fuzzy vectorial space (TFVS) which is described in our previous work [32]. The personality and interaction layers are implemented with a multi-agent assisted decision support (MAADS) [33]. The emotion layer is designed with the Ortony Clore and Collins model (OCC) [34], the personality and interaction layers are implemented with a multi-agent aided decision support (MAADS) [33], the knowledge layer by the knowledge model (KM) that relies on a knowledge classification presented in Section 3.3.4 and the experience layer with the case-based reasoning approach (CBR) [20,33,35–37] described in Section 3.3.5.

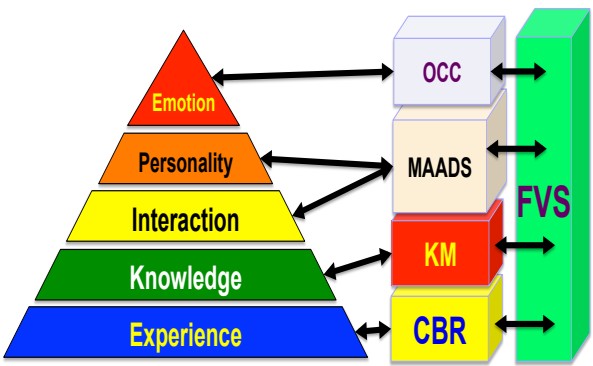

**Figure 9.** The EPICE Model.

### 3.3.1. The Emotion Layer E

According to Picard [38], a complete emotional system must integrate the following components:

- "Emotional behaviour" (time scale expressed in minutes),
- "Fast primary emotions" (time scale experessed in seconds),
- "Cognitively generated emotions" like cognitive dissonance (time scale expressed in minutes or hours),
- "Emotional experience (which is expressed in cognitive awareness, physiological awareness and subjective feelings)",
- Body mind Interactions (time scale is the day or more).

Some emotion models were proposed like Roseman's model [39] and the OCC (Ortony Clore Collins) cognitive model which is intended to represent emotions and their consequences for mood, reflexive (emotional loops) or on positive or negative abilities to act and make skilful decisions [34]. The emotional layer E is composed of attributes that express balance and define the agent's emotional states. The Roseman's model [39] is similar. The layer E model defines a set of 11 emotional parameters that represent both positive and negative intensities Figure 10. Each emotion can change rapidly over time. The implementation of emotion models are based on formal rules and thus makes it difficult to manage time, the choice of status change thresholds and the associated transition functions like in the Flame system [6,40]. Virtual personalities were also proposed in Reference [41]. Most of them are based on fuzzy rules and the emotion models are not well suited to take time into account during the evolution of the situation and the environment.

| | | F(x) | + | - |
|---|---|---|---|---|
| Consequences of events occuring in the environment | For others | f1 | Happy for | Resentement |
| | | f2 | Gloating | Pity |
| | For self | f3 | Hope | Fear |
| | | f4 | Joy | Distress |
| Actions of Agents | Self Agent | f5 | Pride | Shame |
| | | f6 | Gratification | Remorse |
| | Other Agent | f7 | Gratitude | Anger |
| | | f8 | Admiration | Reproach |
| | | f9 | Gratification | Remorse |
| Aspects of Objects | | f10 | Gratitude | Anger |
| | | f11 | Love | Hate |

**Figure 10.** Parameters of the Ortony Clore Collins (OCC) model and functions [34].

For example, if we use the components of the OCC model to implement the emotional state of an individual with a TFVS, the resultant vector $\bigoplus_{f_i=1}^{f_i=11} f_i(x,t)$ represents the emotion state of the individual at each time of the evolution of the modelled situation when they are relevant.

3.3.2. The Personality Layer P

Pathology in decision-making and decision-making processes affects many areas of human activity. The decision (more precisely, the decision-making process) is an entity with blurred contours for which it is not possible to determine a kind of "standard" with respect to which could be detected deviations that would be pathological. Each individual's personality is built throughout his or her life and each individual is unique. Rather, the objective is to define an approach that identifies how mental pathology or even simple traits affect a decision-making process at different levels of space and time granularity. The psychological structure of each person is built during the course of life, first in childhood as highlighted by Jean Piaget [42,43] and then during the important period from adolescence to adulthood. It is a slow, systemic process, as the poet Antonio Machado in "Proverbios y Cantares, 1912" says : Wanderer, your footsteps are the road and nothing more; wanderer, there is no road, the road is made by walking. Walking makes the road, and turning to look behind you see the path that you will never tread again. Wanderer, there is no road, only foam trails on the sea" [44]. In the same way, many authors agree that the life makes the personality and the psychological structure of a person. For example, Boris Cyrulnik proposed the concept of *"resilience"* to explain how people can find in themself the strength to heal their damaged life [45]. Pathologically, two major classes are distinguished:

- **Psychotic pathology** has attributes and characteristics relating to the functioning of the psychism, which are interesting to consider from the perspective chosen here. Let us quote:
  - The alteration of the relationship to reality ranging from cognitive errors (multiple and of very variable importance) to the delirium constituted involving a reconstruction of reality. A decision-making process based on a delusional ideation is likely to lead to a delirious, or at least inadequate, outcome.
  - Disorders of the formal organization of thought (we can include here most of the psychotic defense mechanisms), affecting the logical capacity will leave their mark also. It should be noted, however, that certain delusions (particularly certain purely paranoid constructions) are not accompanied by any alteration in a formal capacity and that, on the basis of false and/or delusional assumptions, decisions can be drawn up which retain all the appearances of the most rigorous and effective logic. Here, despite appearances, can manifest a massive pathology. A seminal work in psychosis modelling and simulation is artificial paranoia [46].
- **Neurotic pathology** also affects the decision—the symptomatology of obsessive neurosis very often involves an inability to stop a decision. In the field of hysteria, the needs of "appearing" are often such in the pathology register that decisions are less important because of their actual content and effects than because of what appears to be involved. Distortions of a nature different from those of delirium can be considerable and certainly pathological.
- **Constitutional factors** like personality and temperament may well be marked by pathology. There are certainly, within the limits of normal, daring temperaments that are prone to quick decisions, as there are cautious temperaments that take them less easily. But even if it is very difficult to draw a clear boundary between the normal and the pathological, at the approaches of extremes, the pathology is manifest and indisputable. Character disorders where they are of some importance, uncontrolled impulsivity can give rise to decisional commitments that are clearly pathological. On the other hand, too poorly constituted objectality can lead to inhibition of engagement, which impairs the ability to make decisions in an awkward way. Finally, let us quote, the very considerable scope of perversion and perverse functioning as determining and often highly pathological elements affecting all decision-making processes. During the decision process we are influenced by the mood and emotions (stress) as well as traits or character structures conducive to or unfavourable to decision-making. The time scale of constitutional factors is expressed in years and they often remain permanent during the whole life or evolve very slowly.

During his life, brutal events or experiences may modify the individual's psychism as described in Figure 11.

Figure 11 describes how new objects are perceived in the environment and can match (or not) with memorized objects in past events and provokes an impact on the emotion and mental state of the person. This impact influences his/her behaviour and his/her decisions. In turn, these consequences and results of these decisions are memorized with the corresponding objects and thus slowly modify in his/her psychological structure in a long term learning loop. Moreover, appropriate and useful behaviors in the environment can be stored in the individual's genetic code and transmitted to his offspring in a very long-term philogenetic loop. For example, in the savannah, the necks of giraffes have lengthened over a long time to reach the ever higher foliage of the trees. Rodents have an intuitive fear of the snakes that are their predators. This zoophobia would have a very ancient origin in the brain of vertebrates and especially mammals.

Figure 12 shows that a neurotic personality relies on a list of anxiogenous events that causes a list of emotional unpleasant effects and that leads the subject to use a list of defence behaviors.

Figure 13 shows three types of neurosis—obsessional, phobia and hysterical neurosis and their corresponding lists of anxiogeneous events, emotion effects and defence behaviors. The model proposes generic prototypes describing typical psychological structures. These prototypes are specialized and thus adapted to the different actors in order to model the behaviours actually observed. They correspond to decision-making patterns that are likely to evolve over an individual's lifetime.

The personality layer is defined by a library of automatons modeling general behaviours and strategies. These automatons define metaknowledge in psychology. They model the decision-making modes corresponding to the evolution of an actor's emotional states. The personality layer is in close relation to the emotional layer which allows us to represent sympathy, antipathy, aversion, self-esteem. The shape of automatons and the functions of transitions allow us to represent particular psychological behaviors. For example, the necessary tasks to choose and buy an object are listed and numbered in the Table 2.

An optimistic actor behaviour can be represented by the automaton Figure 14a, while an obsessive actor will be represented by an automaton that has a great number of feedback loops due to his checking behavior before getting satisfaction as depicted in Figure 14b. The transitions represented by arrows ($+/-$) are triggered or not according to the value of a fuzzy membershipfunctions that compute the level of satisfaction of the actor. The automata are presented Figure 14.

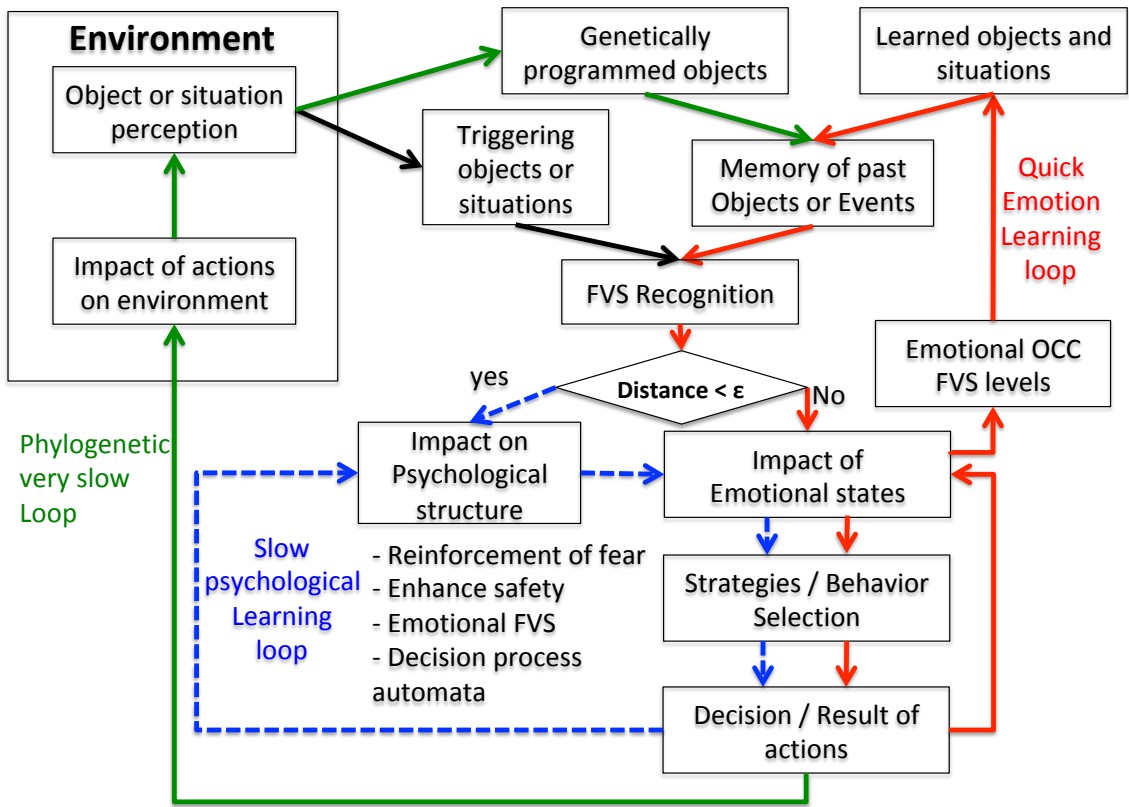

**Figure 11.** The psychological learning loop [47].

**Table 2.** Necessary tasks to choose an object to buy.

| i | Task to Choose an Object to Buy |
|---|---|
| 1 | Set a list of preferences concerning the object |
| 2 | Select a set of objects showing some relevant properties in a group of available objects |
| 3 | Get the specific features of selected objects |
| 4 | Compare the features of the selected objects with the list of preferences |
| 5 | Rank the objects by prices and relevancy |
| 6 | Verify the availability and the warranty of top ranked objects |
| 7 | Decide what object to buy |
| 8 | Pay for the object and get the invoice |

| | | |
|---|---|---|
| **- A list of anxiogeneous events** | | |
| | - An anxiogeneous event is : | |
| | Or | - The perception of a triggering object(s) having specific characteristics (eg. fear of mice) |
| | | - The perception of situation(s) or scanario(s) defining specific contexts (eg. agoraphobia) |
| **- A list of emotional effects** | | |
| | - An emotional effect is : | |
| | And | - The increase or a decrease of emotional parameters |
| | | - The Emotional states should trigger defence behaviors |
| **- A list of defence behaviors** | | |
| | And | - The agent sets a goal (most of the time : the elimination of the unpleasing situation or of the object causing the disagreeable emotional state |
| | | - The agent elaborates a defence strategy. Each strategy is a set of connected actions executed to achieve the goal (destroy the object, flight, avoid, prevent…). |
| | Or | - The agent may show inappropriate behaviors caused by the pathological neurotic personality |

**Figure 12.** A generic model of neurotic personality [47].

| Neurosis | Triggering object | Anxiogeneous events | Emotional effects | Defence behavior |
|---|---|---|---|---|
| Obsessional neurosis | Lack of objects<br><br>Lack of misdoing | Anxiety of loosing an object.<br>Anxiety of not achieving appropriate tasks | Severe inner struggle<br><br>Fear and anxiety | Ritual and necessity to carry out actions again and again Compulsive need to verify |
| Phobia neurosis | Animals<br><br>Crowd<br><br>Closed places | Abnormally intense dread of certain objets or specific situations | Dread<br><br>Fear | Flight Avoiding Prevent |
| Hysterical neurosis | Herself or himself image<br><br>Others' sight | Anxiety of offending or displeasing | Anxiety<br><br>Halluci–nations | Repression Flight Avoiding indifference Versatility Theatrical behavior |

**Figure 13.** A simplified neurosis classification [47].

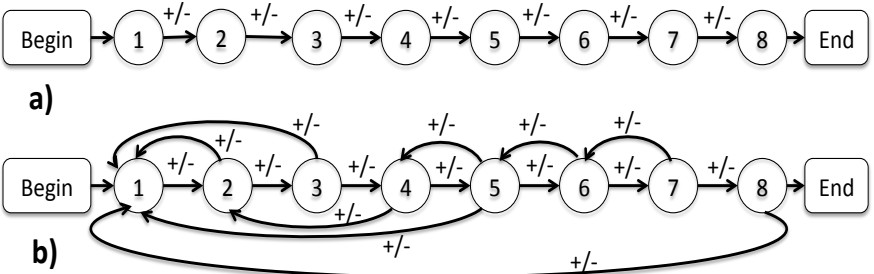

**Figure 14.** Automata describing (**a**) Optimistic personality (**b**) Obsessional personality.

### 3.3.3. The Interaction Layer I

The interaction layer I describes the relationships and interactions that an actor has with other actors in the system. The complexity is linked to the diversity of the actors involved. Each actor can maintain relations with both legal persons and natural persons. The question that arises is what is the place of the various actors within a system of decision-making support in ethics in the context of humanitarian health. Catherine Fuchs in her book revisits the hypothesis formulated by Sapir and Whorf of linguistic relativism [48,49]. For Edward Sapir, language reflects a community's vision and interpretation [50]. His work is based on the classification of Native American languages. For Whorf there is no objective or universal reality, but only representations of it that would be determined by language. Catherine Fuchs explains that each language constructs a different "world view" because each language community selects distinct isolates of experiences and gives them shared meaning. Regarding the question of variants and invariants (one-sided): "the diversity of representations constructed through languages is a central question for linguistics: the theory of articulation concerning variations and invariants."

We think that the triptych Language, Thought and Cognition is in action here—according to Roman Jakobson, thought differs according to language. For Pinker and Fodor there is no thought without language. Is there a thought and cognition without language? Do we think the same way as the Japanese or the Lapps? Louis Léry clearly showed that he was not. The culture of a people is organized around verbal and non-verbal modes of communication, beliefs, custom describes all the areas and constraints that will be involved in the decision. It also shows the hierarchy of decision-making criteria: law, ethics, charters, customs, beliefs and religion [51]. The brain is like a plant that grows in an environment. Our knowledge is linked to our culture and our environment. Can it be transferred to other people of different cultures? It is doubtless possible, if one spends long years in a country, to acquire the customs and customs and to forget, unlearn one's previous culture, which will gradually fade away. This is just a hunch based on observation of a few cases.

Our person can develop only in interrelation with others, hence the importance of the «relational flower». How do we model interactions between legal and physical persons or groups of persons? A n-m relationship is necessary in order to represent all the interactions that may exist between many actors. The graph complexity increases dramatically as $(na.nr)^2$ where na is the number of actors and nr is the number of modeled relationships between them.

Models from the work of linguists such as Quillian on semantics have been reused and adapted by computer scientists [52] J.F. Le Ny shows the interest of semantic networks [53]. The ontology models are intended for the axiomatisation of a field of knowledge [20].

### 3.3.4. The Knowledge Layer K (Connaissance C)

The knowledge layer defines the sources of knowledge in which an actor has access because of his roles and of his skills. We define five categories of knowledge in an independent way chosen models of representation and domains of knowledge.

- Factual knowledge: it is about data describing an object of the world real and generally admitted by all. The observed facts are confidentially connected to the truth and classified according to their degree of certainty and precision. It is about a statistical argument—for example, the majority of people have a similar perception of a characteristic of an object, for example its color is red.
- Knowledge heuristics: if a situation S is then observed we have knowledge which are relevant and valid in this situation S. It can be properties of objects or a usually applicable method successfully in this situation. The causes of the validity of a knowledge heuristics are not always available.
- Procedural knowledge: how it acts on the world—know the chains of tasks to be made to reach an expected result. It is about all the procedures or the courses to follow expressed by a list of tasks realized to be effective in a given situation.
- Dynamic or behavioral knowledge: they concern the spontaneous variation of the facts, or a behavior in time which is usually observed (for example, the earth rotates around its axis once

every 24 h). They are useful for the simulation of natural phenomena. It is about the perception of the various states spontaneously taken by objects during given period, of their interactions. The behavioral knowledge have a major importance in sociology, in economy, in botany, in medicine and in physics where we observe the behavior of a system.
- Deep learning is the new type of knowledge that can be now implemented in systems. System knowledge modelling relies on operators detecting the similarity of knowledge objects.

Other important aspects are time modeling and the reasoning operators: deduction, induction, abduction, subsumption and analogy that allow to combine and to compare knowledge objects.

- Deduction is based on the modus ponens or modus tollens.
- Induction tries to propagate a property observed in one object to all objects that belong to the same class and allow to split the class in two subclasses the one with the property and the other without the property.
- On the contrary, abduction tries to refute a property usually oberved in the objects of a class.
- Subsumption tries to generalise properties to more general concepts—think of the individual under the general (an individual under a species, a species under a genus); consider a fact as understood under a law. General assumption could be applied to implement induction as proposed by Buntine in 1988 [54].
- Analogy transposes the relationships and properties of objects from one universe to another one provided that these universes and object classes are sufficiently similar. The similarity is computed by a distance as in case-based reasoning approach.

### 3.3.5. The Experience Layer E

The experience allows to enhance the knowledge in a specific area of the science. Case-based reasoning (CBR) implements a kind of analogy. CBR is a model of experience that allows us to index and store cases in an object-oriented database and then, in next consultation, to use a distance to retrieve in the database the more similar cases to the new case in order to apply the most appropriate methods of the previously stored cases to solve the new problem [35–37]. CBR involves semantic distances developed by different approaches—algorithms of structural similarity [55]; statistical learning as proposed by [56]; digital approaches from neural networks and fuzzy logic. The distances are using to implements the different stages of the CBR cycle as depicted on Figure 15 [57]. The research on semantic distances tends to combine symbolic and numerical aspects [20].

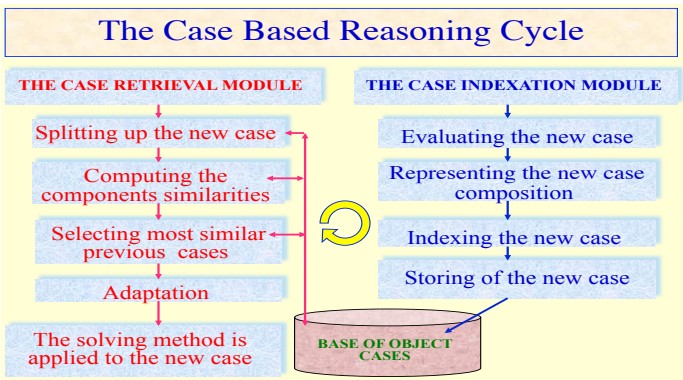

**Figure 15.** A Case Based Reasoning Cycle [33].

### 3.4. An Artificial Thinking Model (ATM)

For linguists such as Pinker and his pupil Fodor, there is no thought without language, that is without words indicating the objects of the world. Jerry Fodor proposes an internal language named the "mentalese" which gives to each a reflexive thought, the capacity to speak to oneself and so to repeat for one of the information useful for his/her life. This reflexive thought is strongly bound to

the consciousness of oneself and his/her body, to exist, to be an alive entity of the world with his history, souvenirs, feelings and projects. Even there, the self-awareness: "The I" appears late in the life of the individual, during childhood and even in adolescence. The capacity to think is previous to the implementation of the language. When we speak to oneself, the internal verbalization stimulates the same intellectual cortical zones as during the expression by the word with request of the driving ways, but the motricity of apophyses aryténoïdes which tighten the vocal cords would be inhibited by the brain preventing the emission of sounds of the voice [58]. Thus there it would be no important difference to speak and speak to himself (in the sense of the mentalais of Fodor), in both cases the areas of the language are requested Figure 16. We speak to ourselves to strengthen our capacity of analysis and resolution of the problems but this activity is not necessary for thought and do not even maybe fatal in a perception more lit of the world. Does speaking to oneself constitute the only way of thinking?

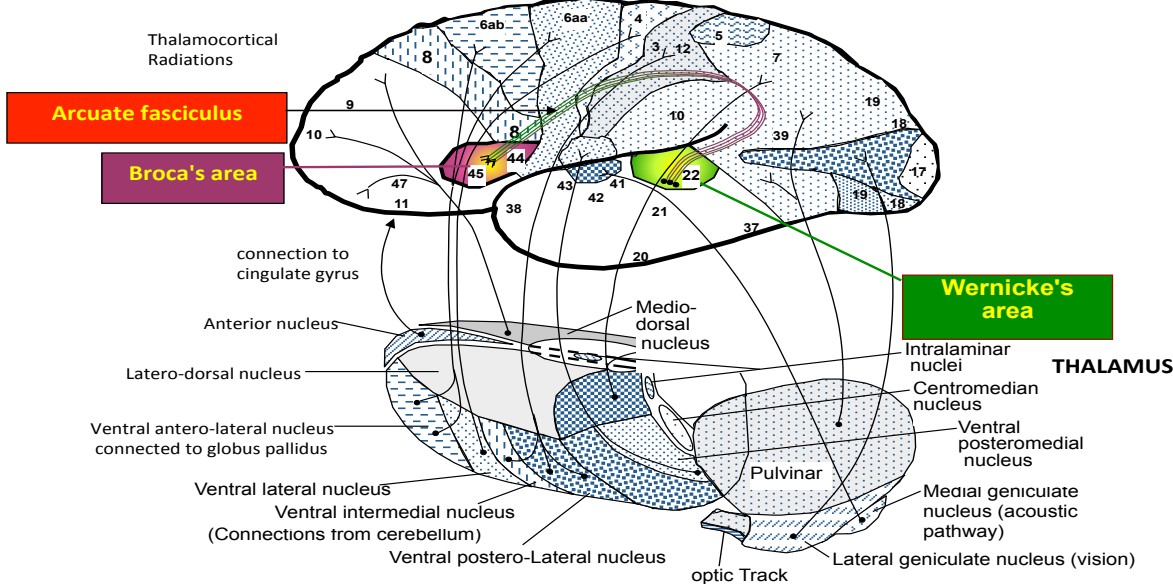

**Figure 16.** The brain, language area and thalamic connections [1,58].

Symbolic and Subsymbolic Way of Thinking

Computers could think in a very different way to humans, as a plane flies differently to but faster than a bird [59]. The Zen is a total holism, the world cannot be absolutely divided into parts. The dilemma is that for every object of the world, according to master Zen Mummon, "we cannot express him with words and we cannot express him without the words". According to Buddhism, to trust words to reach the truth is equivalent to trusting an always incomplete formal system [60]. For Jiddu Krishnamurti, our consciousness is common to all humanity—all human beings thinking contributes to building it. He considers the individualism, the ego as an obstacle to understand the consciousness with rare moments of clarity ("insight") [61]. *"The thought is a movement in the time and the space. The thought is memory, memory of the past things. The thought is the activity of the knowledge, the knowledge which was gathered through millions of years and stored in the form of memory in the brain."* [61]. There are two forms of thinking: The first is a reaction of the memory which contains the knowledge, the result of the experience from the beginning of humanity (phylogenesis) and since our birth (epigenesis) in a loop: experience knowledge memory thought action and so on, necessarily limited by the time, it is used every day, rational, individualistic, power-hungry and of progress submissive to the knowledge which accumulates, in the words which divide and this division is responsible for all the suffering, for all the troubles of the world.This first type of thinking relies on the symbolic treatment of the information and the linguistic hierarchy upper right part of

Figure 17 with linguistic operators and able to implement the mentalese as described by Jerry Fodor. The alternative thinking—*the "active-attention" that occurs in rare occasions, we simply pay our attention to the world, without interpreting it, without naming anything, virgin to any prejudice, knowledge and especially spontaneously, by living this moment without thinking of it and without the will. For example, the direct perception of a wonderful landscape of a mountain one morning with all our senses uses our complete attention where we forget ourselves and banish the use of the words"* [61]. Such a full perception requests only the left lower part: the sub-symbolic cognitive pyramid and not the linguistic pyramid Figure 17.

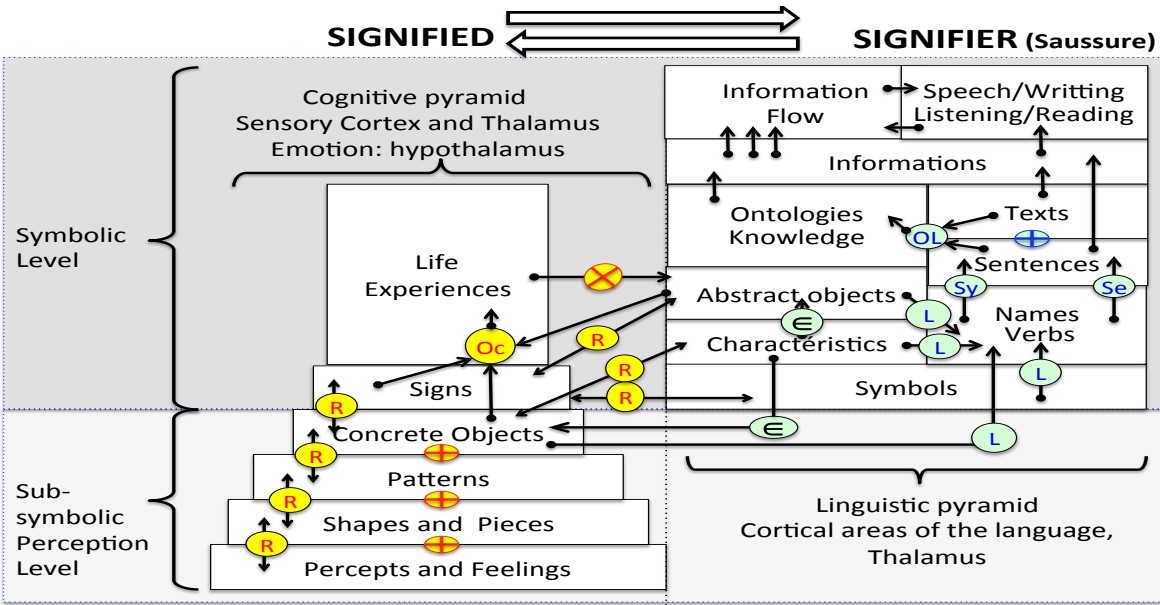

**Figure 17.** The cognitive pyramid [1].

Vincent Lamareille psychanalyst and information system designer has described the psychoanalysis according to Lacan with UML (Unified Modeling Language) diagrams [62]. Figure 18 describes the signified and Figure 19 shows the signifier: *"Language gives the subject representation by a significant, it is alienation. But this is not enough for life, ex-sistence implies significant interaction, intersignificance, social bond."*. Lacan was inspired by the works of Ferdinand de Saussure but on the contrary he performs a separation between signifier and signified [63].

The first way of thinking is a fatal vicious circle implemented soon in computers that will think better than human beings! And nothing prevents a computer inventing a new religion at the origin of new sufferings for humanity. Language during its evolution would have made man lose his spontaneity in the immediate perception of the world such as it is. Digital technology, which strengthens the symbolic nature of our relation to the world, is doubtless going to establish the peak of our ignorance. The realization of CAMAS being able to think is now possible. Alain Cardon proposes a psychic system that relies on knowledge in psychology and that is capable of generating flows of thoughts that take place in the temporality under the shape of organized groups of processes to build the artificial psychic system and its interactions. This conscious autonomous multi-agent system (CAMAS) is able to choose its objectives and set its goals to achieve them (level 9 of the Le Moigne's classification of complex systems. Alain Cardon recently published a method to develop systems with psychological states and with the symbolic way of thinking inspired by Sigmund Freud [64,65].

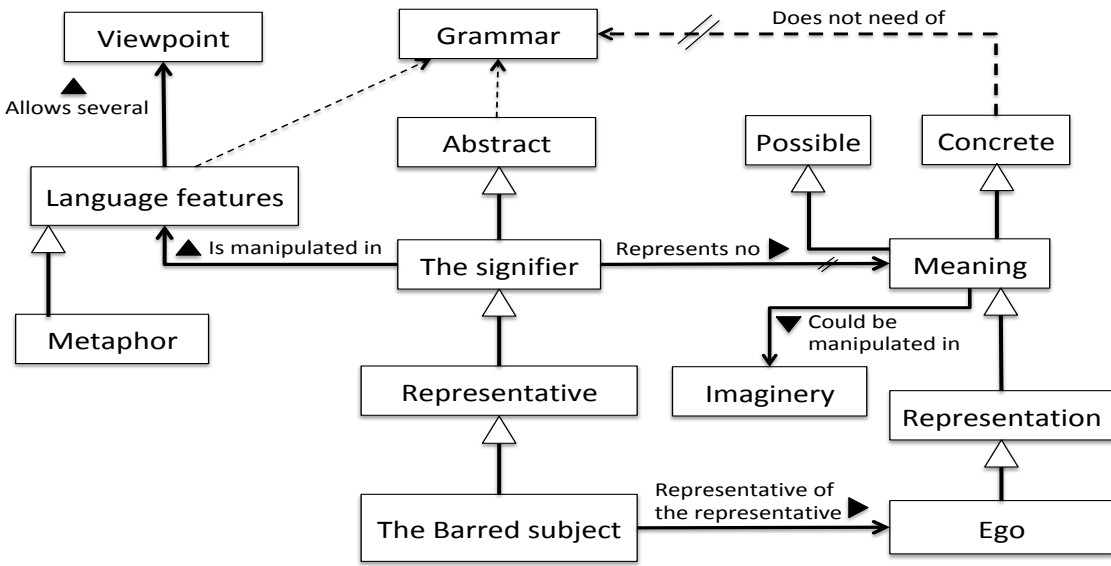

**Figure 18.** "The signified" Unified Modeling Language (UML) diagram Tr. into english from (Lamareille 2009) [62].

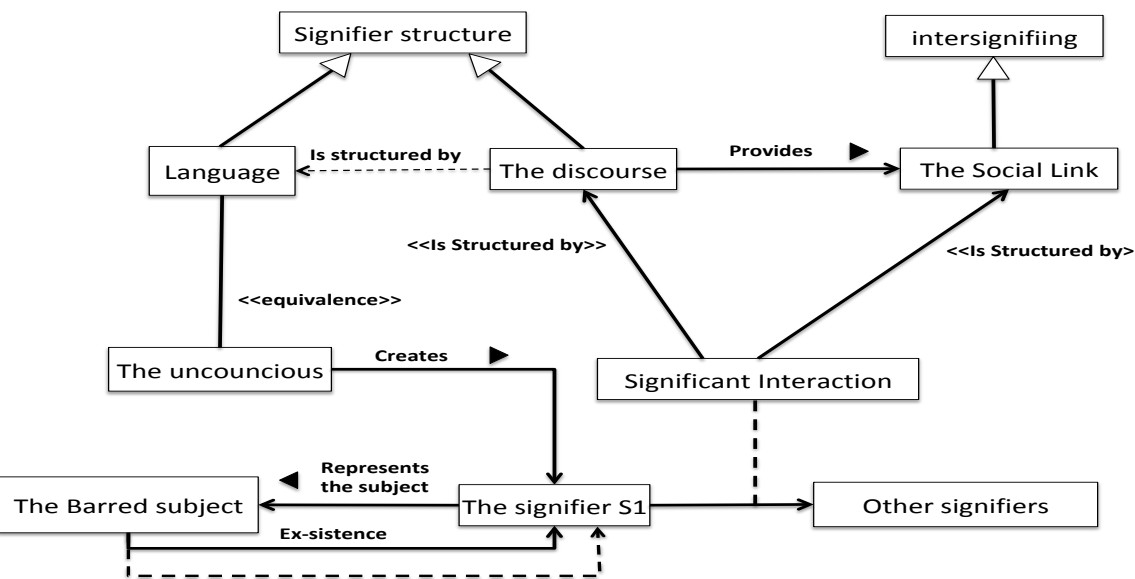

**Figure 19.** "The Signifier" UML diagram Tr. into english from (Lamareille 2009) [62].

The CAMAS can be reproduced in very large numbers and coordinate each other to reach complex goals fixed in common. The CAMAS is thus going to overtake the human being in his capacity to be thought and to act in the world. Furthermore, it is endowed with a mechanical, robotics architecture allowing it to move and to act in the world with more power and efficiency than human beings. It can be endowed with artificial senses like ordinary sight but also infrared vision, large spectral hearing, nano-cameras amplifying its perception far beyond our possibilities. The main advantage would be to be able to improve the capacities of investigation of our world including in hostile circles as the conquest of space by taking advantage of the concept of telepresence without risk [66].

These systems have access to the Internet and will quickly become more powerful than human beings with disturbing consequences for the future of the humanity. We have shown that computers can think in the first way only. The main characteristic of humanity is to still be able to think also in the alternative way but how long will it be before we lose our essential faculty.

## 4. Results

### 4.1. Application to Cognitive Dissonance and Decision Making

The cognitive dissonance concept was proposed by Leon Festinger in 1957. 'Cognitive dissonance could be defined as an internal tension in the person's thoughts that concern a mismatch between their beliefs, emotions cognitions and attitudes when several of them contradict each other. The cognitive dissonance concerns also the uneasiness a person experiences when a behaviour conflicts with their ideas or beliefs (stored in his memory). Festingers studied the strategies to reduce induced psychological tension maintain personal consistency, including strategies to avoid circumstances identified as a source of dissonance [67,68]. The concept of dissonance cognitive was widely used in marketing to model customer behaviour and allow sellers to leverage it to influence their purchasing decisions. We think that it is interesting to use the OCC model to describe cognitive dissonance. The "Franklin effect" is a cognitive dissonance situation where Benjamin Franklin had to face an opponent and wrote him to solve the situation: *"This is another instance of the truth of an old maxim I had learned, which says,"He that has once done you a kindness will be more ready to do you another, than he whom you yourself have obliged." And it shows how much more profitable it is prudently to remove, than to resent, return, and continue inimical proceedings".* B. Franlin autobiography # 121- # 122, pages 72–73 [69]. The imprinting which is a latent learning that occurs when the subject is placed in a situation without reward or punishment. It occurs willingly with the help of visual stimuli and concerns a precise object that is valued, attractive, becomes sought after and creates an affective attachment [70]. Anchoring is a type of imprinting that causes the perception of cognitive dissonance because it creates a contradiction with the previous perception of an object perceived as idealized. This cognitive dissonance is exploited in marketing as in the following scenario:

-   Step 1: At the entrance of an antique store a pretty object is exhibited and proposed at a high price x (for example 5000€).
-   Step 2: The customer continues to visit the store and discovers objects of no interest to him at average prices.
-   Step 3: further in the same store you will find a similar object but half price (2500€) than the previous one. Obviously, the first perception of this object at a very high price leads you to think that it is the usual price for such an object and that the second object is indeed very cheap and you rush to enjoy the bargain and there the trap closes. In fact, the first object was very overvalued and so is the second.You thought you were making a good deal but in fact the real value of the object was only 1000€ and you were fooled.

### 4.2. TFVS Modelling of Emotion in Buying Decision

The graph on Figure 20 shows the object model with the emotions relations according the OCC model and fuzzy membership functions $f_1 to f_{11}$ described in the emotion layer of the EPICE model defined Section 3.3.1, Figure 10. We use the sigmoid fuzzy membership functions defined in Figures 7, 8 and 21 to respectively describe the emotion felt by the customer (likes/dislikes), his assessement concerning the displayed price of the object (cheap/expensive) and the perceived value of the object (poorValue/greatValue).

Figure 7 shows the price membership function (where $-1$ is to be cheap and 1 is to be expensive) according to the price in € and according to the customer's budget which is 2600€ and the membership function Figure 8 describes the value of the object perceived by the customer 1: great value (100%) and $-1$: very poor value (0%). The higher the value perceived by the customer (false or not) the more likely the customer is to buy the product at a high price.

According to the situation, the relevant couple of parameters of the OCC model can be chosen from those listed in the table OCC presented Figure 10, Section 3.3.1. Each couple of OCC parameters can be similarly represented by a couple of membership functions and then transformed in the corresponding vectors as depicted on Figure 22.

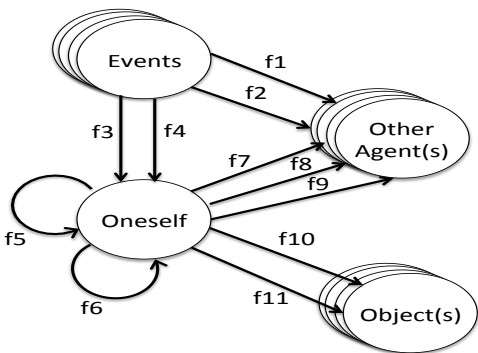

**Figure 20.** A graph of the emotion relation according to the OCC model.

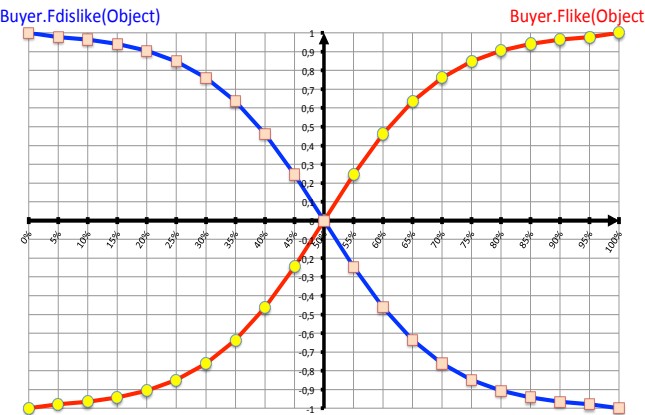

**Figure 21.** Like Membership functions according to OCC *fLikes() fDislikes()*.

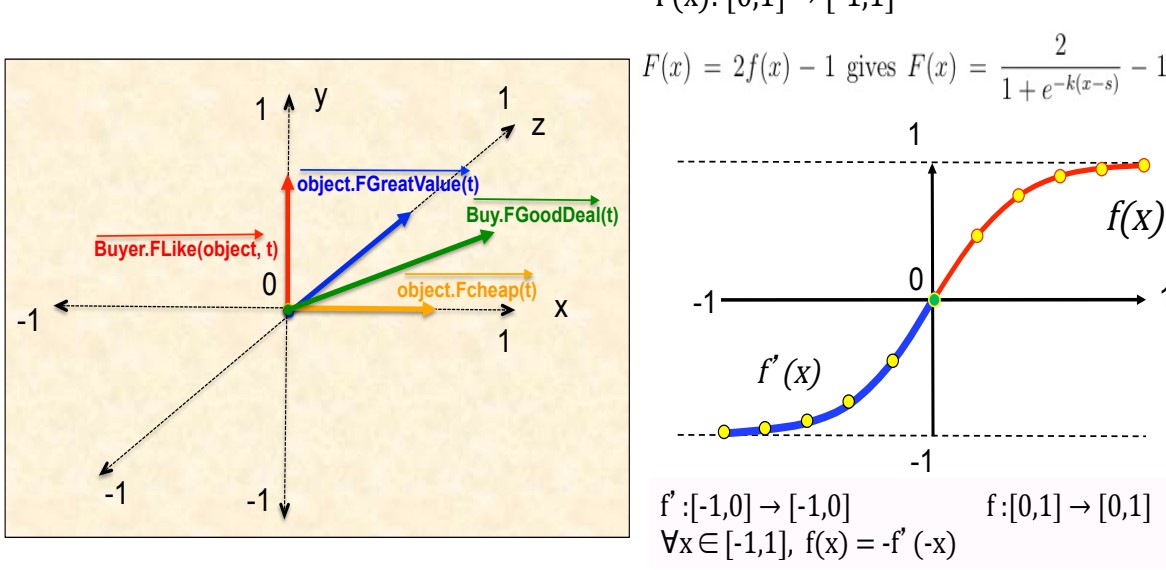

**Figure 22.** Time-fuzzy vector space (TFVS) of a good deal indicator.

A good buy indicator is to buy a good value object that I like at a cheap price:

$$\overrightarrow{buy.FGoodDeal(t)} = \overrightarrow{buyer.Flike(object, t)} + \overrightarrow{object.Fcheap(t)} + \overrightarrow{object.FgreatValue(t)}. \quad (11)$$

The decision to buy (or not) is depicted in Figure 23 and is rather straightforward. The figure shows the evolution of the situation of cognitive dissonance described in Section 4.1 and how to model it with the OCC model and a TFVS. The customer keeps in mind the perception of object 1 and object n is of the same type as the first object and in fact they are not. The advantage of the seller is that he knows the true value of the object he is selling and the fair price.

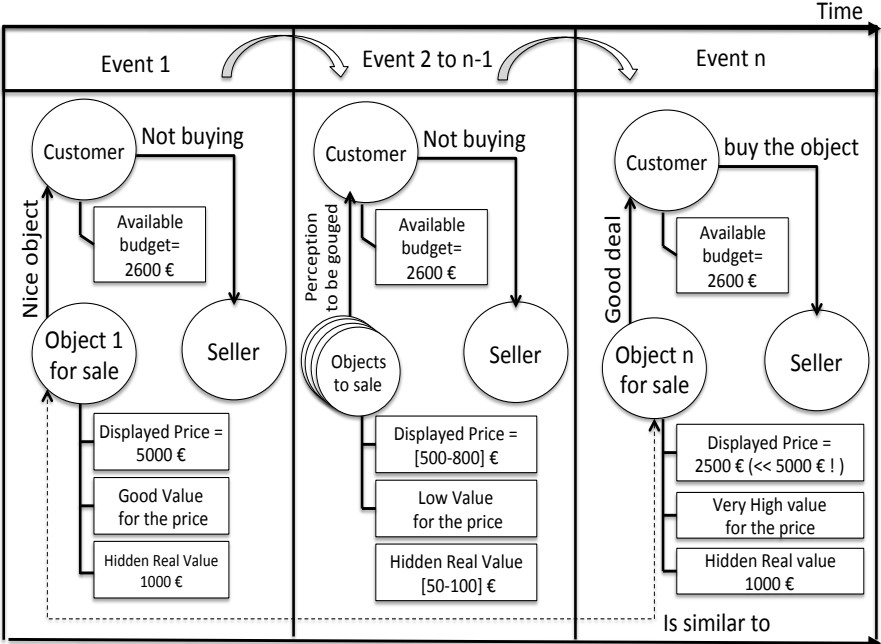

**Figure 23.** A graph of the emotion relation according to the OCC model.

### 4.3. Game Addiction Application

Another example of cognitive dissonance concerns game addiction, the evolution of the emotion during gambling and the impact of a win or loss on the emotions of the player and thus on their way of playing during a game session. When the player begins to lose, he or she will experience an internal conflict between two attitudes—continue to play and hope a big win (but a big amount of money is already lost) or to stop with the fear of somebody else sitting down at the machine and winning the jackpot in your place.

4.3.1. The Game Addiction Model

The game addiction model example needs to describe the interaction between a player and a slot machine during the game and his/her emotion evolution. Another example shows how the model is used to implement the decision of the player when he is gambling and if he wins or not. This application is presented in our previous work [32].

Figure 24 shows a slot machine and the buttons that the gamer can use to bet and that are described by the automata in Figures 25–27.

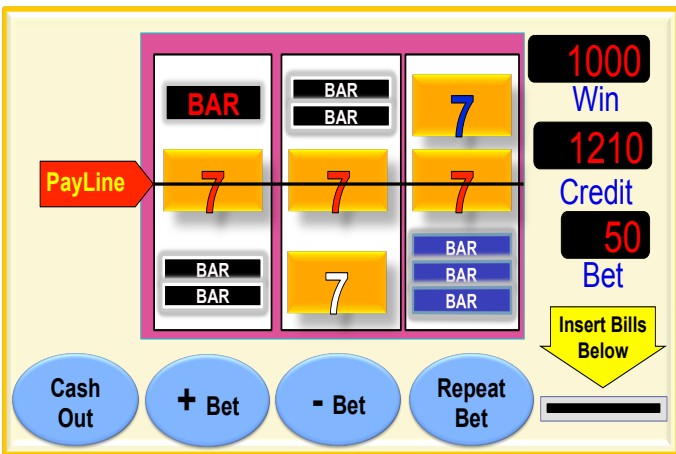

**Figure 24.** Example of slot machine.

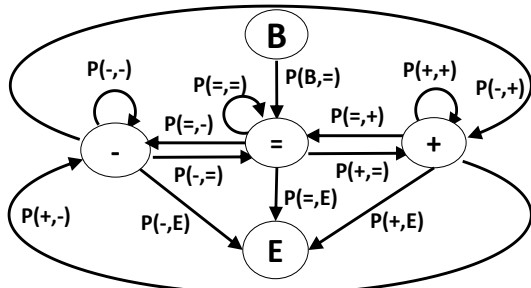

**Figure 25.** Automaton defining the possible game actions of the player

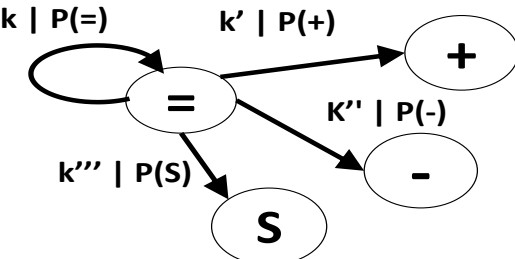

**Figure 26.** Automaton defining the next bet

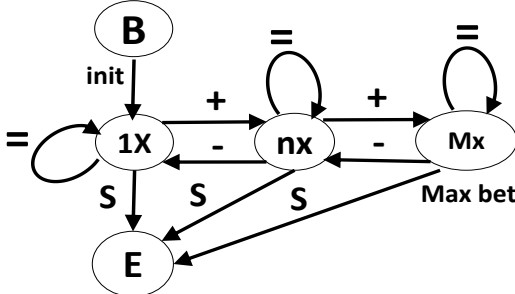

**Figure 27.** Automaton describing the available transitions of the slot machine

The model uses finite-state automata with multiplicities composed of a set of internal states characterizing the mental situation of the player and transitions from one state to another (see Figures 25 and 26). The Figure 27 describes the automaton of the slot machine and its different states and transitions. In the following, the events k are evaluated by the score in the interval $[-1,1]$ by a function E in Equation (12). Each transition is associated with a probability that indicates the

chance to occur shown on Figure 26. They are mutually exclusive and the sum is 1. The automaton behavior at each time t is defined by a 5-uple $A(t) = (\Sigma, Q, I, J, TFVS)$. $Q$ is the finite set of internal behavior states, $\Sigma$ is the set of perceived events k evaluated by a score in the interval $[-1, 1]$. $I$ and $J$ are respectively the set of initial and final states of transitions. As in the previous example, a TFVS is defined by several pairs of opposite fuzzy membership functions, three in this example: f: Joy, f′: Distress; g: Hope, g′: Fear; and h: Pride, h′: Shame, and their corresponding vectors $\vec{F}(t)$, $\vec{G}(t)$ and $\vec{H}(t)$ according to Section 3.1 and the OCC model (Figure 20). After each bet at time t, the value of the resultant vector $\vec{U}(t)$ that defines the mood of the player is calculated.

$\forall (u, v) \in I \times J :$

$$X : \begin{cases} P(u,v) \in [0,1] E : \Sigma \to I \times J \\ \forall k \in \Sigma, P(u,v) = E(k) \\ if \ x \leq P(u,v) \ then \ X = u \\ else \ X = v. \end{cases} \tag{12}$$

### 4.3.2. Decision of the Next Bet

During the game, the player decides what to do next with the next bet automaton Figure 26 where the $P(=)$, $P(+)$, $P(-)$ and $P(S)$ are evaluated by the function E according to the emotion provoked by the events $k \in \Sigma$ (win, lose or special events like free games etc.) that occurred in the previous game. If a win occurs, the player is willing to increase the bet with the hope of future other wins and thus P(+) is incremented while P(−) and P(=) are decremented. In this place, the psychology and strategies of the player should be taken into account and the automaton in Figure 25 shows the possible decisions of the player during two successive bets and their corresponding values in Table 3. The automaton on Figure 26 determines what could be the next bet and their corresponding probabilities according to player's mood. The output of the mapping is a stochastic value determined by a random variable X described in equation 12. This variable determines the bet decision of the player. According to the player action, the slot machine is thrown and gives the result of the bet. And so on until the end of the game: state E of Figure 27 where the slot machine print a ticket with the amount of money available.

**Table 3.** Transition Values.

| Transition | Value | Signification |
|---|---|---|
| P(B,=) | 100% | Always initial state |
| P(=,=) | 25% | Stable, neutral state |
| P(=,+) | 25% | Raise, moderate optimism |
| P(=,−) | 30% | decrease, moderate pessimism |
| P(=,E) | 20% | Cashout discouragement, loss |
| P(+,=) | 60% | Decrease, wait for a bright spell |
| P(+,+) | 10% | Double increase, euphoria |
| P(+,−) | 5% | Brutal decrease, fear, pessimism |
| P(+,E) | 5% | Brutal cashout: loss |
| P(−,=) | 20% | End of pessimism, moderate attitude |
| P(−,+) | 10% | From decrease to increase, optimism |
| P(−,−) | 15% | Double decrease, annoyance, anger |
| P(−,E) | 12% | Cashout discouragement, loss |

The retroaction loop of emotion/decision makes the player simulator able to take appropriate decisions based on the previous results of the game that may provoke emotional consequences.

## 5. Discussion

This paper is an extended version of a conference paper, ESM'2019 EUROSIS [1] where a prospective of AI towards autonomous systems was presented. It is also inspired by our work

concerning time modeling and reasoning which are very important in clinical decision support systems. Dealing with time is mandatory to appropriately design the behavior of diseases and their evolution. The object-oriented time model was proposed to describe the clinical pictures of acute drug intoxications in Reference [27].

Previous works presented ontologies agents and the experience layer, the case based reasoning approach in medicine [20,21,71,72].

We firstly present the TFVS model and then the new method FVSOOMM to implement the layers of the EPICE model which provides thinking features inspired by the human being. We have presented the different layers of EPICE that are based on psychological knowledge of cognitive dissonance and the OCC model of emotion. This complex framework is necessary to simulate the artificial thinking behavior of human beings shown in Figure 17. The cognitive and linguistic pyramids both rely on the composition object-oriented relationship in Figure 17 presented in Reference [1]. The TFVS model provides means to implement the personality and the emotion layers and interaction layers of EPICE on which relies the decision process implemented with automata. We have shown two examples of cognitive dissonance—the buying decision of an object and the game addiction application. These examples show how the events in the environment and the results of previous acts are influencing the mood and the emotions and thus the willing of the next bet decisions of the agent whose behavior is similar to that of a human gamer.

The proposed approach is based on a time-fuzzy vector space (TFVS) which is an extension of an object-oriented model initially based on the object composition [73]. This approach applies fuzzy characteristic functions on object composition (is-part-of) and object-attribute relationship (has-a) whose existence is Boolean (exists or does not exist) in classic object models. Thus our approach is not of fuzzy logic, but it is an extension oriented object. It is therefore totally compatible with the fuzzy logic of Zadeh, whose operators can moreover be combined to implant deduction and induction especially on the characteristic functions of attributes of the domain of knowledge but also, in the form of vectors with encapsulated attributes of object model objects, but also (and this is new) to combine these vectors to specify the states of simple objects and complex objects as a result of composition relationships.

*Future Work*

We are also using the model in clinical medicine where taking into account the time modeling is mandatory to properly design the process of doing the diagnosis and to choose the treatment of evolving diseases. However, at the moment, the UML diagram ontology, the membership functions, the TFVS are designed by hand and requires a close collaboration between the system designer and the domain expert. Each case is modelled on a spreadsheet with the various fuzzy features needed. A java application has been developed to build from the objects stored in a database, the fuzzy membership functions and the corresponding fuzzy vectors. We are currently developing a design support platform to manage the steps of the FVSOOMM method from the domain ontology class diagram to the EVF model and to test the resultant vectors at each level of the composition hierachy.

The platform should facilitate the realization of agents with personality and emotions according to the EPICE model and allow to compare their behavior over time in similar situations and thus to realize convincing open simulators of the mental behavior.

We are also studying how to implement and combine the object analogy with the different reasoning modes (induction, abduction, subsumption) to enrich the decision process.

That will allow us to define a temporal case-based reasoning approach based on a dynamic distance able to compare complex compound objects over a specific time interval. Such a platform facilitates the uptake of the model by the end-users and the development of new applications in many areas. Another exciting current work is how to implement insight capabilities using our TFVS model.

Another important track of our research is AI ethics where the advantages and drawbacks of the use of AI systems must be carefully observed and evaluated. However to know what AI can already do

help to foresee what could be developed tomorrow—new features and their benefits and drawbacks. The goal of our work is not to go beyond the human intelligence but to evaluate what is possible to simulate and what remains the prerogative of the human. We think that most knowledge and reasoning processes can be simulated by AI. On one hand we recognise that this can be useful because of transferring advice and know-how and enhancing education. On the other hand, we agree with Krishnamurty that the essence of human intelligence and thought lies in a direct apprehension of the environment the *"active attention"* that constitutes the original way of perception of the world based not on logic, language and symbolic thinking that allows only a partial and reductionist approach.

## 6. Conclusions

This work relies on a previous paper presented at "Modelling and Simulation 2019 (ESM'2019) [1]. This extension presented four contributions—the TFVS model which is a fuzzy object-oriented model that allows us to design the values of the attributes and the behavior of the object of the system over the time. The method FVSOOMM that describes the necessary steps to design the TFVS from a UML class diagram representing the domain ontology, the EPICE cognitive model whose layers describes the necessary objects to design the artificial thinking model in which architecture is presented as the fourth contribution. The results present two examples of cognitive dissonance to illustrate our approach. They show how to model emotion and environmental events in order to influence the decision which is implemented with finite states automata whose transitions are dynamically selected according to the resultant vector.

The two first contributions—TFVS and FVSOOMM—are important contributions in computer sciences because they provide a framework for time modeling in many domains where the fuzzy object-oriented approaches can be used and when a dynamic behavior and interactions between objects must be taken into account. For example we are currently developing an application of our TFVS model on territorial policy for urban communities in our region in order to understand and explain how mayors make their decisions and power issues in urban communities. The psychological behavior of leaders must be considered. The EPICE and ATM models can be used in robotics to simulate human thinking and behavior with the ethical limits associated with the achievements of such avatars. We are using TFVS to represent the evolution speed, acceleration and bifurcation of the flow of thoughts. Artificial intelligence (AI) can be useful to us if and only if it can preserve a scrupulous respect for the identity and freedom of end-users. End-users should keep the power over the AI system, the choice of using them or not and therefore should remain a tool of help and not be transformed into subjects to which undesirable and unattended services could be provided at any moment, as is the case with some help agents in word processing that want to change what I type in a very inappropriate manner.

## 7. Patents

EVF (FVS) model and methods are registred in "l'agence des dépots numériques".

**Funding:** This research received no external funding.

**Conflicts of Interest:** The author declares no conflict of interest.

## Abbreviations

The following abbreviations are used in this manuscript:

| | |
|---|---|
| AI | Artificial Intelligence |
| ATM | Artificial Thinking Model |
| CBR | Case-Based Reasoning |
| EVF | Espace Vectoriel Flou |
| FVS | Fuzzy Vector Space |
| FVSOOMM | Fuzzy Vector Space Model and Method |

KADS    Knowledge Analysis and Design Support
KOD     Knowledge Oriented Design
MASK    Méthode d'Analyse et de Structuration des (K)Connaissances
OCC     Ortony Clore and Collins model
TFVS    Time Fuzzy Vectorial Space
UML     Unified Modeling Language

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
