# Peer review of "Fvsoomm a Fuzzy Vectorial Space Model and Method of Personality, Cognitive Dissonance and Emotion in Decision Makingâ€"

_information, doi:10.3390/info11040229_

Round 1
Reviewer 1 Report
Dear editor,
Thank you for sending me for review the paper “FVSOOMM A FUZZY VECTORIAL SPACE MODEL OF PERSONALITY, COGNITIVE DISSONANCE AND EMOTION IN DECISION MAKING”.
General comments: The paper is really impressive for the efforts made from you to demonstrate the valence of your method. The method, well explained by the use of the math, is clear and the results are good. In my opinion the paper has potential for publication in Information. However, the author(s) need to consider the following points as limitation or further scope for refining the paper.
- All abbreviations should be introduced in abstract and then in the main text, for example, EPICE, UML, AI etc.
- Abstract should be rewritten. Please, add next parts in the abstract: Purpose, Design/methodology/approach, Findings, Practical implications and Originality.
- Page 1, line 20, authors wrote “This work is an extension of the proposed article at the ESM’2019 conference” Reference for this conference paper is missing.
- Introduction section: 1) Please clearly summarise what specific advantages brings your approach. 2) Try to specific that and build a case for your research (focus on novelty). This should be presented in at least one paragraph. 3) Please clarify what is the novelty of your work. 4) Please provide aim of this paper. 5) Highlight the novelty of study in the introduction.
- Literature review is missing. You should discuss papers related to application of AI algorithms in decision making. Based on that discussion you should define gap that your approach is filling up.
- Section 2 is too long. You have 19 subsections, this is too much. This section should be separated into two different sections. The first part is theoretical and that part should be shorten/reduced. The second part is mathematical and that part should be in present from along with examples that are explaining your mathematical formulations.
- Results are good, but should be discussed in more effective way. Section 3 is very important for validation of your algorithm and should be presented in more effective way. Can you propose to the readers comparison with other existing approaches? This will help you to validate your approach.
- Section 4 should be removed in the first part of the paper.
- Conclusion- Add more future scope. Summarize your results. Also, how the proposed method can be applicable to other real life problems need to be mentioned. Do not use bullets or numerations in this section.
Author Response
Reviewer 1
Thank you for sending me for review the paper “FVSOOMM A FUZZY VECTORIAL SPACE MODEL OF PERSONALITY, COGNITIVE DISSONANCE AND EMOTION IN DECISION MAKING”.
General comments: The paper is really impressive for the efforts made from you to demonstrate the valence of your method. The method, well explained by the use of the math, is clear and the results are good. In my opinion the paper has potential for publication in Information. However, the author(s) need to consider the following points as limitation or further scope for refining the paper.
Dear reviewer, first of all I would like to thank you for the time you have given to review my article and for your valuable comments that should make it significantly better. Thank you for your kind compliments and for your valuable advice to improve this article.
- All abbreviations should be introduced in abstract and then in the main text, for example, EPICE, UML, AI etc.
It’s true I forgot to define the first time they are used some of the abbreviations used in the article. It is now done. I also identified as requested the acronyms used at the end of the article in the abbreviations section
- Abstract should be rewritten. Please, add next parts in the abstract: Purpose, Design/methodology/approach, Findings, Practical implications and Originality.
Thank you for that very thoughtful suggestion. So I completely rewrote the summary, taking into account the plan you proposed by defining the acronyms that were used the first time they were met and introducing the parts that you recommend purpose, design methodology and approach (4 contributions) Findings : how they cooperate and interact together to implement the artificial thinking model (ATM) , the practical implications : the design and implementation of the two example (buying decision and gaming addiction) and at last originality of the approach to design the elements of the ATM.
- Page 1, line 20, authors wrote “This work is an extension of the proposed article at the ESM’2019 conference” Reference for this conference paper is missing.
I add the reference of the ESM’2019 conference paper as the first reference [1] .
- Introduction section: 1) Please clearly summarise what specific advantages brings your approach. 2) Try to specific that and build a case for your research (focus on novelty). This should be presented in at least one paragraph. 3) Please clarify what is the novelty of your work. 4) Please provide aim of this paper. 5) Highlight the novelty of study in the introduction.
Since I completely restructured the article by completely changing the plan, which required many changes to accommodate your remarks about « Section 2 being too long and Section 4 being moved back to the first part of the article ». The first part presents in turn the four main contributions of the article with their advantages and limits.
So I completely rewrote the introduction by respecting the points you mentioned and introducing these new parts of the article.
I started with a review of the state of the art as requested to show the specifics of our TFVS approach compared to the classic fuzzy logic.
In particular, it is a matter of showing that our approach is simpler and that it allows the implementation of the principles of fuzzy not only on variables using logical operators but a new fuzzy object-oriented model that implements the fuzzy paradigm on attributes and object composition with vectors.
I then introduced the plan by articulating the four main original contributions presented in this article: the TFVS models, the FVSOOMM method that guides the designer of the design from the ontology in UML to the realization of the vectors of the TFVS.
I described the originality of the artificial thought model (ATM) and the notion of cognitive dissonance and how to model it with our model. I then illustrated this approach by presenting the interactions with two examples of applications the modelling of the anchor case in the sales decision and the evolution vectors during the visit of the shop and the addiction to games on slot machines. This example must be reused to search on the addiction process that leads people to be very addicted playing and even they are loosing they play and play again...
Thank you for considering my results as interesting and so I followed your advice by explaining more precisely the validation of the algorithm and in particular by adding explanatory diagrams figures 18 to 22 that show how the modelling and the representation in the form of fuzzy vectors.
The discussion identified the benefits and limitations of our approach. It ends with a part that presents our future work including the development of a reasoning based on temporal cases and a platform implanting the FVSOOMM method and facilitating the use of our model.
Comparison with existing approaches takes place in the state of the art and then in the course of the water in the different layers of EPICE, for example the model FLAME in section 3.3.1.
The limits of our approach is also discussed and especially because thinking is much beyond of the linguistic approach of what thinking is, this aspect was more presented in the conference paper ESM’2019. This article is more focused on the design and implementation purpose and the feasability
- Literature review is missing. You should discuss papers related to application of AI algorithms in decision making. Based on that discussion you should define gap that your approach is filling up.
Thank you, I agree, for the fuzzy logic approach I introduced a new section : « State of the art of knowledge and time modeling with fuzzy logic ».
For the cognitive aspects, the state of art concerning (psychological, linguistic domains) is included in the description of the EPICE model and the ATM model to remains them near the place they are used.
- Section 2 is too long. You have 19 subsections, this is too much. This section should be separated into two different sections. The first part is theoretical and that part should be shorten/reduced. The second part is mathematical and that part should be in present from along with examples that are explaining your mathematical formulations.
Thank you very much for the positive assessment of the results that are now presented in section 4 and I add figures to explain more deeply my approach and the examples. The comparison with existing approaches are mainly done in the new state of art section at the beginning of the paper.
- Results are good, but should be discussed in more effective way. Section 3 is very important for validation of your algorithm and should be presented in more effective way. Can you propose to the readers comparison with other existing approaches? This will help you to validate your approach.
Thank you very much for the assessment of the results that are now presented in section 4 and was completed to explain more deeply my approach. The comparison with existing approach are mainly done in the new state of art section at the beginning of the paper.
- Section 4 should be removed in the first part of the paper.
That is true, thank you very much to suggest that, I completely redesigned the plan according to your very relevant suggestions. This section becomes the second contribution in the first part of the paper
- Conclusion- Add more future scope. Summarize your results. Also, how the proposed method can be applicable to other real life problems need to be mentioned. Do not use bullets or numerations in this section.
I agree, the conclusion was far too brief and did not sufficiently present the contributions and perspectives of this article. I have added a paragraph on future work in the discussion and also on the work that is currently being done with the urban community.
Reviewer 2 Report
Review report for the paper “FVSOOMM A FUZZY VECTORIAL SPACE MODEL OF PERSONALITY, COGNITIVE DISSONANCE AND EMOTION IN DECISION MAKING”.
The authors should revise the paper according below listed comments:
- This paper presents expanded idea from conference paper. You should add the conference paper in the reference list.
- Some information in section 2 are redundant. You should better organize this section.
- Use abbreviations in a proper way. When you use abbreviation for very first time, you should explain it.
- Figure 25 should be presented before case study. Also, case study should follow flowchart organization.
- Can we use triangle or trapezoidal fuzzy functions in the model? Why you have used only sigmoid fuzzy function?
- Discussion section should be revised.
- Conclusion section seems like rush to the end. Add limitation/advantages of proposed approach. Better present future directions.
Author Response
Reviewer 2
Review report for the paper “FVSOOMM A FUZZY VECTORIAL SPACE MODEL OF PERSONALITY, COGNITIVE DISSONANCE AND EMOTION IN DECISION MAKING”.
The authors should revise the paper according below listed comments:
Dear reviewer, first of all I would like to thank you for the time you have given to review my article and for your valuable comments that should make it significantly better.
- This paper presents expanded idea from conference paper. You should add the conference paper in the reference list.
Thank you, that is true, I mentionned the fact but I have forgotten to add the reference of the paper. I did add the reference of the ESM’2019 conference paper as the first reference [1] .
- Some information in section 2 are redundant. You should better organize this section.
It is true, I thank you for this remark, Section 2 was far too long and I completely reworked the plan in order to better present the 4 contributions. In order to avoid redundancies, I grouped together the elements that talked about the same subject in different parts of the article.
- Use abbreviations in a proper way. When you use abbreviation for very first time, you should explain it.
It’s true I forgot to define the first time they are used some of the abbreviations used in the article. It is now done. I also identified as requested the acronyms used at the end of the article in the abbreviations section
- Figure 25 should be presented before case study. Also, case study should follow flowchart organization.
That’s true, but since I’ve reworked the article plan, figure 25, which was at the end of the article, becomes figure 6 in section 3.2. which presents the FVSOOMM method as a second contribution and which is now before the presentation of the cases in the results section.
- Can we use triangle or trapezoidal fuzzy functions in the model? Why you have used only sigmoid fuzzy function?
Yes it is possible to use triangular and trapezoidal characteristic functions to represent the values of an attribute. Our approach is not fuzzy logic but a fuzzy object oriented model, it allows to represent discrete attributes or whose evolution of values is triangular or trapezoidal. Sigmoid functions are used because they become vectors that are combined with a-one relationships and the composition of objects. This standardization of our approach makes it possible to give it a certain generality and to extend it to a large number of fields of application.
Our approach is not fuzzy logic but a fuzzy object oriented model, it allows to represent discrete attributes or whose evolution of values is triangular or trapezoidal.This standardization of our approach makes it possible to give it a certain generality and to extend it to a large number of fields of application.
- Discussion section should be revised.
I agree, the discussion section has been completely reworked in particular to reflect the change in plan.
- Conclusion section seems like rush to the end. Add limitation/advantages of proposed approach. Better present future directions.
I agree, the conclusion was far too brief and did not sufficiently present the contributions and perspectives of this article. I have added a paragraph on future work in the discussion and also on the work that is currently being done with the urban community.
Round 2
Reviewer 1 Report
I am very happy that the authors have addressed my concerns point by point precisely. No further suggestions come from my side. Therefore, I would like to recommend this manuscript to be published.
Reviewer 2 Report
All the reviewers' comments have been addressed carefully and sufficiently, the revisions are rational from my point of view, I think the current version of the paper can be accepted.